# SToFM: a Multi-scale Foundation Model for Spatial Transcriptomics

**Suyuan Zhao** [1 2]  **Yizhen Luo** [1 2 3]  **Ganbo Yang** [2]  **Yan Zhong** [4]  **Hao Zhou** [1]  **Zaiqing Nie** [† 1 3]

## Abstract

Spatial Transcriptomics (ST) technologies provide biologists with rich insights into single-cell biology by preserving spatial context of cells. Building foundational models for ST can significantly enhance the analysis of vast and complex data sources, unlocking new perspectives on the intricacies of biological tissues. However, modeling ST data is inherently challenging due to the need to extract multi-scale information from tissue slices containing vast numbers of cells. This process requires integrating macro-scale tissue morphology, micro-scale cellular microenvironment, and gene-scale gene expression profile. To address this challenge, we propose **SToFM**, a multi-scale **S**patial **T**ranscript**o**mics **F**oundation **M**odel. SToFM first performs multi-scale information extraction on each ST slice, to construct a set of ST sub-slices that aggregate macro-, micro- and gene-scale information. Then an SE(2) Transformer is used to obtain high-quality cell representations from the sub-slices. Additionally, we construct **SToCorpus-88M**, the largest high-resolution spatial transcriptomics corpus for pretraining. SToFM achieves outstanding performance on a variety of downstream tasks, such as tissue region semantic segmentation and cell type annotation, demonstrating its comprehensive understanding of ST data through capturing and integrating multi-scale information.

## 1. Introduction

In single-cell transcriptomics, single-cell RNA sequencing (scRNA-seq) data reflects the gene expression levels in

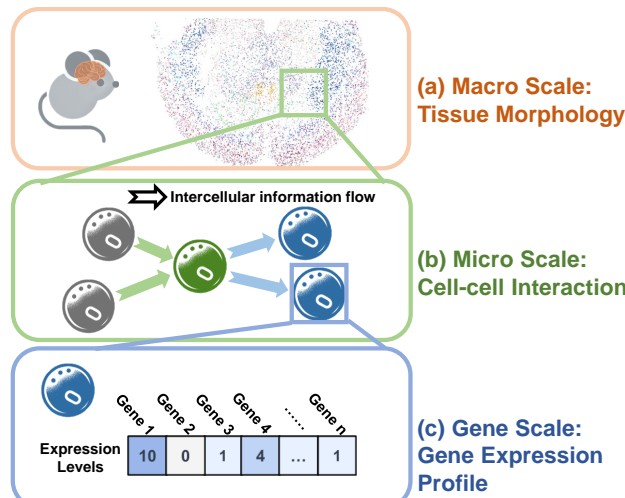

*Figure 1:* **ST data contain biological information from multiple scales. (a).** An example of a visualized ST slice of a mouse brain. **(b).** A microscopic sub-slice of the brain slice. **(c).** Gene expression profile of a single cell in the brain slice.

cells, providing a digital representation of cells (Ziegenhain et al., 2017). Machine learning approaches have achieved great success in analyzing scRNA-seq data and enhancing the understanding of complex biological systems (Szałata et al., 2024). However, scRNA-seq technology disassociates cells from their tissue contexts, and therefore overlooks tissue-level information such as tissue morphology and intercellular relationships. This limitation compromises the application scope of scRNA-seq data in real-world studies involving cells. (Williams et al., 2022; Du et al., 2023).

Fortunately, Spatial Transcriptomics (ST) technology has opened up new possibilities to measure gene expression in cells without losing spatial arrangements of tissues. Specifically, ST data are obtained in the form of tissue slices and can be computationally described as a 2D point cloud containing tens of thousands of points. Each of these points represents a cell or a sequencing spot[1] containing up to approximately 20,000 gene expression values. Such rich information in ST data offers biologists a unique perspective on single-cell biology (Du et al., 2023; Marx, 2021). Effec-

---

[†]Corresponding author  [1]Institute for AI Industry Research (AIR), Tsinghua University [2]Department of Computer Science and Tecnology, Tsinghua University [3]PharMolix Inc. [4]School of Mathematical Sciences, Peking University. Correspondence to: Suyuan Zhao <sxdtzsy@gmail.com>, Zaiqing Nie <zaiqing@air.tsinghua.edu.cn>.

*Proceedings of the 42$^{nd}$ International Conference on Machine Learning*, Vancouver, Canada. PMLR 267, 2025. Copyright 2025 by the author(s).

[1]For simplicity, we refer to both cells and sequencing spots as "cells" in the remaining paper since we only use high-resolution ST data close to single-cell resolution. See Appendix D.1 for further explanations.

tively analyzing ST data can provide cell-level insights for various tissue structure-related studies, such as embryology (Chen et al., 2022), neuroscience (Shi et al., 2023; Hasel et al., 2021), tumor microenvironment analysis (Wu et al., 2021; Jin et al., 2024), and disease studies related to specific organs (Wu et al., 2024; Franzén et al., 2024).

The large amount of ST data generated in recent years pose growing demand for analysis (Atta & Fan, 2021), which has promoted the application of machine learning methods in this field. Early attempts design specific approaches for various ST downstream tasks, such as deconvolution, imputation and clustering (Biancalani et al., 2021; Abdelaal et al., 2020; Dong & Zhang, 2022). More recently, inspired by the success of pretrained models in single-cell transcriptomics (Yang et al., 2022; Cui et al., 2023; Hao et al., 2023; Theodoris et al., 2023), some research efforts have attempted to develop ST foundation models. For example, Nicheformer pretrains a Transformer-based model on single-cell gene expression profiles from both ST data and scRNA-seq data for transferability (Schaar et al., 2024). CellPLM further integrates spatial information by encoding cell coordinates with sinusoidal position embeddings (Wen et al., 2023). These models demonstrate superior performance and transferability across a series of downstream tasks.

Despite the promising advancements, we argue that one of the most important characteristics of ST data has not been well captured by existing works. As illustrated in Fig. 1, ST data contain biological information from **multiple scales**. From a *macro scale* (Fig. 1a), we can extract tissue morphology and organ structure information such as functional zones and anatomical layers. From a *micro scale* (Fig. 1b), we can capture cellular contexts and cell-cell interactions by analyzing intercellular relationships with spatially adjacent cells. From a *gene scale* (Fig. 1c), we can delve into detailed information for each cell by analyzing gene expression profiles. Analyzing ST data requires a comprehensive understanding of biological information across all the different scales. For example, in order to comprehend brain structure and function in neuroscience, macro-scale information like brain anatomical regions, micro-scale information like neuron connections, and gene-scale information like spatially variable genes are all indispensable (Jung & Kim, 2023). This presents a unique challenge for establishing an ST foundation model: capturing and integrating multi-scale information from a large number of ST slices with appropriate model architecture and self-supervised objectives.

To address this challenge, we propose **SToFM**, a multi-scale ST foundation model, which is shown in Fig. 2. Firstly, to capture *gene-scale* information, SToFM performs domain adaptation to an off-the-shelf pretrained cell encoder (Theodoris et al., 2023) by continual pretraining on gene

expression profiles from ST data. Secondly, to capture *micro-scale* information, we divide a ST slice into multiple ST sub-slices based on the cell coordinates, and use an SE(2) Transformer (Zhou et al., 2023) to capture spatial information and cell-cell interactions. This division not only emphasizes the spatially localized cell-cell interactions (Zormpas et al., 2023) but also improves the computational efficiency in fully harvesting ST data. Finally, to maintain *macro-scale* information within each sub-slice, we draw insights from multi-scale approaches in CV (Cao et al., 2019; Liu et al., 2021) and GNNs (Ying et al., 2018; Xu et al., 2018) and represent macro-scale information as a few virtual cells through clustering, and inject them into the sub-slices. We design two pretraining tasks: masked cell modeling and pairwise distance recovery, to capture both gene expressions and spatial characteristics of ST data.

To train SToFM, we construct **SToCorpus-88M**, the largest high-resolution ST pretraining corpus to date. This corpus includes approximately 2,000 high-resolution ST slices obtained by 6 different ST technologies, totaling 88 million cells. It surpasses the current largest ST corpus (Schaar et al., 2024) by 1.5 fold in scale and 2 fold in ST technology diversity. SToCorpus-88M will be publicly released.

In order to validate the effectiveness of SToFM in integrating multi-scale information from ST data, we establish a comprehensive benchmark containing several significant biological tasks. Specifically, SToFM surpasses the best baseline by 11.6% and 14.34% on $F_1$ score in the cross-slice tissue region semantic segmentation task (Chan et al., 2019; He et al., 2024) and the cell type annotation task (Yang et al., 2022; Domínguez Conde et al., 2022), respectively. We further justify the informativeness and transferability of SToFM representations through additional qualitative and quantitative results. We attribute this to the integration of multi-scale information, since tissue morphology, cell-cell interaction patterns and gene expression semantics all contribute to obtaining consistent and transferable representations across different ST slices.

Our main contributions are summarized as follows:

(1) We propose SToFM, a multi-scale foundation model that captures and integrates information from macro, micro and gene scale of spatial transcriptomics.

(2) We construct SToCorpus-88M, the first large-scale spatial transcriptomics corpus suitable for pretraining. SToCorpus-88M will be publicly available.

(3) We demonstrate the outstanding performance of SToFM on various downstream tasks involving ST data.

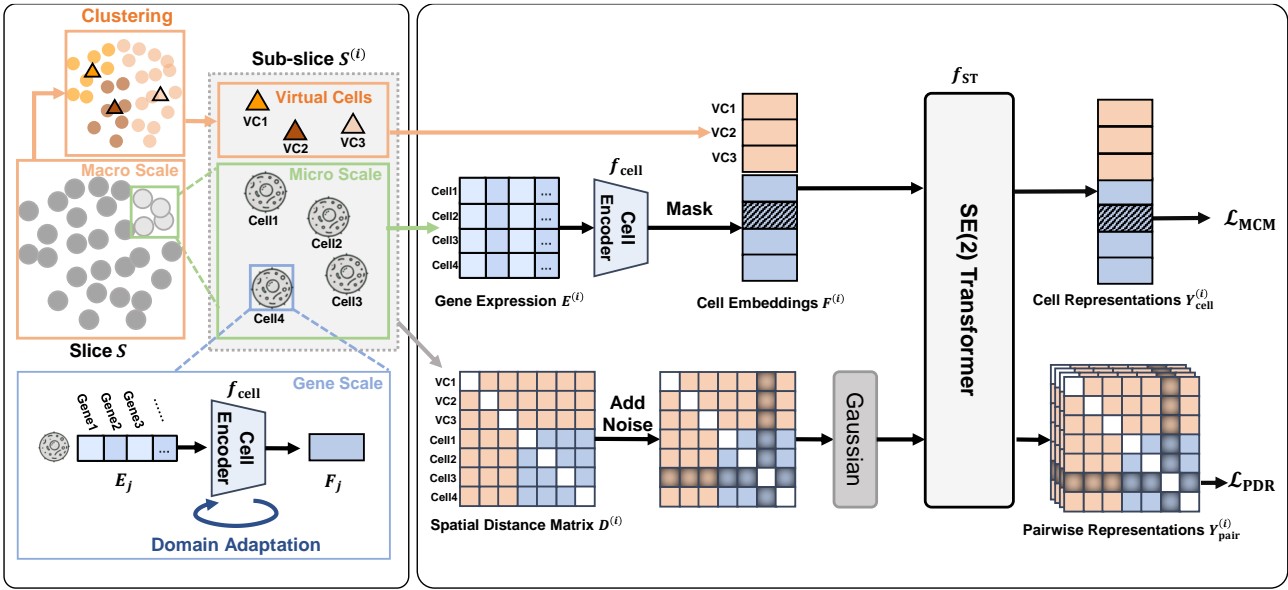

*Figure 2:* **An overview of SToFM architecture. (a).** Multi-scale processing integrates information on macro, micro, and gene scales into sub-slices. At gene scale, we extract the representations of each cell using a cell encoder that has undergone domain adaptation. At micro scale, we divide the ST slice into several sub-slices. At macro scale, we cluster cells based on spatial and gene expression data, construct virtual cells for each cluster, and add the virtual cells into each sub-slice. *Notably*, The actual sub-slice contains a much larger number of cells (~1000) and virtual cells (~50) than are shown in the simplified figure. **(b).** During the pretraining process, first, the domain-adapted cell encoder is used to re-calculate cell embeddings, enabling the cell encoder to be trained through backpropagation. Next, an SE(2) Transformer is employed to jointly model the randomly masked cell embeddings and the noise-augmented distance matrix. The output representations are then used to reconstruct the original cell embeddings and spatial distances. These two pretraining tasks are referred to as masked cell modeling (MCM) and pairwise distance recovery (PDR).

## 2. Related Work

**Single-cell Foundation Models** The development of scRNA-seq technology has led to an exponential growth of the data (Svensson et al., 2018), which provides the basis for single-cell foundation models. Innovated by pretrained models in the NLP domain, most existing single-cell foundation models such as scBERT and Geneformer leverage Transformers architecture and language modeling objectives like masked language modeling to capture the inherent gene expression patterns (Yang et al., 2022; Gong et al., 2023; Theodoris et al., 2023; Zhao et al., 2023; Cui et al., 2023; Hao et al., 2023). Other studies further integrate single-cell data with natural language for multi-modal pretraining to expand biological insights (Zhao et al., 2024; Levine et al., 2023; Choi et al., 2024; Yuan et al., 2024). Benefiting from large-scale pretraining, these single-cell foundation models have demonstrated effectiveness on a variety of downstream tasks in single-cell biology (Wenteler et al., 2024; Liu et al., 2023) and provided novel insights to biologists (Bordukova et al., 2024; Yang et al., 2024). We refer readers to (Szałata et al., 2024) for a more comprehensive review.

**Deep Learning Methods for Spatial Transcriptomics** Early machine learning-based methods in ST data analysis primarily focus on specific downstream tasks, employing architectures with small parameter sizes and without large-

scale pretraining. For example, Tangram (Biancalani et al., 2021) effectively performs deconvolution on low-resolution ST data using reference scRNA-seq datasets. STAGATE (Dong & Zhang, 2022) excels in downstream tasks like tissue structure segmentation and spot deconvolution. SpaGE (Abdelaal et al., 2020) integrates ST and scRNA-seq datasets to impute the gene expression profiles of ST data. More recently, developing ST foundation models with large-scale unsupervised ST data has attracted rising research interest. Nicheformer (Schaar et al., 2024) is a Transformer-based model pretrained on single-cell from ST data to mitigate the distributional gap introduced by spatial sequencing technology. It only utilizes gene expressions of ST data and ignores the spatial coordinates. To our best knowledge, CellPLM (Wen et al., 2023) is the only prior work that integrates spatial information with transcriptomics in pretraining by combining gene expression embeddings and position embeddings as the Transformer's input embeddings. As a nascent attempt, CellPLM has preliminarily validated the feasibility of establishing an ST foundation model. In contrast, SToFM captures fine-grained gene expression embeddings at gene scale and also proposes more advanced approaches for incorporating information at micro and macro scales.

# 3. Methods

In this section, we describe the implementation and workflow of SToFM, which is illustrated in Fig. 2. First, we perform multi-scale information extraction to each ST slice, concentrating macro-, micro- and gene-scale information within sub-slices, which are feasible inputs for a neural network to handle, as described in Sec. 3.1. Then, for each sub-slice, we use an SE(2) Transformer to combine transcriptomics and spatial information for representation learning, as described in Sec. 3.2. Finally, in Sec. 3.3, we describe the pretraining objectives and strategies in detail.

## 3.1. Multi-scale Information Extraction

We represent an ST slice containing $N$ cells as: $S_0 = (E, P)$, where $E = \{E_i \in \mathbb{R}^n\}_{i=1}^N$ is the $n$-dimensional gene expression matrix, and $P = \{P_i \in \mathbb{R}^2\}_{i=1}^N$ represents the 2D positional coordinates. As shown in Fig. 2a, in this process, we perform multi-scale information extraction on each slice to obtain a set of sub-slices that aggregate multi-scale information, defined as $\hat{S} = \{S^{(i)}\}$. $\hat{S}$ will be used as model input for the next process.

**Gene Scale Domain Adaptation**  Given the high dimensionality and sparsity of gene expression profiles, we need to obtain their dimensionality reduction embeddings. Due to technical limitations, ST data often exhibit deficiencies such as limited gene coverage and high rates of "dropout zeros" (Avşar & Pir, 2023; Li et al., 2023). Consequently, compared to scRNA-seq data, the quality of ST data is considerably lower. Incrementally training scRNA-seq foundation models on ST data facilitates to transfer the knowledge learned from scRNA-seq data to ST data with different distributions, achieving higher-quality cell embeddings. We initialize our cell encoder with Geneformer, one of the most advanced Transformer-based single-cell foundation models, which achieves high-quality representation by encoding cells as gene sequences arranged in descending order of relative expressions (Theodoris et al., 2023). Please refer to Appendix A.1 for more details.

We incrementally train our cell encoder on the cells from ST data to achieve domain adaptation. The domain-adapted cell encoder $f_{\text{cell}}$ is then used to process the gene expression profiles $E$ of the ST data, resulting in $h$-dimensional features $F = \{F_i \in \mathbb{R}^h\}_{i=1}^N$, where $F_i = f_{\text{cell}}(E_i)$. We add $F$ as additional information of the cells to $S_0$, and obtain $S = (E, F, P)$.

**Micro- and Macro-scale Integration**  We design a multi-scale approach to effectively integrate information from micro scale and macro scale. We divide each slice into multiple sub-slices based on spatial locations $P$, with each sub-slice containing a manageable number of cells (approxi-

---

**Algorithm 1** Micro- and Macro-scale Integration

1: **Input:** $E = \{E_i\}_{i=1}^N$, $F = \{F_i\}_{i=1}^N$, $P = \{P_i\}_{i=1}^N$, $\alpha \in [0,1]$;
2: $F_c \leftarrow \alpha \cdot \text{Normalize}(\text{PCA}_2(F)) + (1 - \alpha) \cdot \text{Normalize}(P)$
3: Clusters $\leftarrow$ Clustering($F_c$)
4: $F^{\text{VC}} \leftarrow \emptyset$, $P^{\text{VC}} \leftarrow \emptyset$
5: **for** $\mathcal{C}$ **in** Clusters **do**
6: $\quad F^{\text{VC}} \leftarrow F^{\text{VC}} \cup \{\sum_{j \in \mathcal{C}} F_j / \|\mathcal{C}\|\}$
7: $\quad P^{\text{VC}} \leftarrow P^{\text{VC}} \cup \{\sum_{j \in \mathcal{C}} P_j / \|\mathcal{C}\|\}$
8: **end for**
9: SubSlices $\leftarrow$ Split($P$)
10: $\hat{S} \leftarrow \emptyset$
11: **for** $\mathcal{T}$ **in** SubSlices **do**
12: $\quad E^{(i)} \leftarrow \{E_j | j \in \mathcal{T}\}$
13: $\quad F^{(i)} \leftarrow \{F_j | j \in \mathcal{T}\} \cup F^{\text{VC}}$
14: $\quad P^{(i)} \leftarrow \{P_j | j \in \mathcal{T}\} \cup P^{\text{VC}}$
15: $\quad \hat{S} \leftarrow \hat{S} \cup \{S^{(i)} = (E^{(i)}, F^{(i)}, P^{(i)})\}$
16: **end for**
17: **Return** $\hat{S}$

---

mately 1,000). This division empirically achieves a trade-off between maintaining computational efficiency and preserving sufficient localized intercellular interactions. Then, to maintain macro-scale information, we use Leiden algorithm (Traag et al., 2019) to cluster all cells in $S$ based on both cell embeddings $F$ and cell positions $P$. We aggregate each cluster into a virtual cell, whose embedding and position coordinates are averaged over all cells in the cluster. These virtual cells retain the main morphology and partition of the slice, serving as a compression of macro-scale information. We incorporate the virtual cells into each sub-slice, allowing the model to learn micro-scale information while maintaining an ability to perceive macro-scale structural organization. As shown in Algorithm 1, we split $S = (E, F, P)$ into a set of sub-slices $\hat{S} = \{S^{(i)} = (E^{(i)}, F^{(i)}, P^{(i)})\}$ with virtual cells. The sub-slices encapsulate micro-scale information while also integrating macro- and gene-scale information through virtual cells and cell embeddings, respectively. Notably, we only calculate the embeddings $F^{\text{VC}}$ instead of the gene expressions for virtual cells, because the average gene expression profile of multiple cells is out of distribution and cannot be effectively encoded by the cell encoder.

## 3.2. Multi-scale ST Representation Learning

The multi-scale ST representation learning process is shown in Fig. 2b. Given a sub-slice $S^{(i)}$ obtained from Sec. 3.1, we obtain cell representations that incorporate transcriptomics and multi-scale spatial information.

**Second-time Forward with the Cell Encoder**   We perform a second-time forward with the cell encoder, so that the parameters of the cell encoder can be updated through gradient backpropagation. This step is only performed during training, and $F^{(i)}$ can be used directly during inference.

**Representation Learning by SE(2) Transformer** SToFM aims to jointly encode the cell embeddings $F^{(i)}$ and the cell positions $P^{(i)}$, to obtain spatial-aware representations and capture intercellular interactions. In addition, the output representations should be invariant to 2D translations and rotations to $P^{(i)}$, which is known as SE(2)-invariance. For this purpose, we use a widely-adopted SE(2) Transformer architecture[2] (Zhou et al., 2023). This simple and efficient architecture has shown excellent performance in representing proteins (Zheng et al., 2024) and small molecules (Zhou et al., 2023).

Specifically, a distance matrix $D^{(i)}$ is used as input for location information, where $D_{jk}^{(i)} = \|P_j^{(i)} - P_k^{(i)}\|_2$. The distance matrix is first passed through a Gaussian module (Shuaibi et al., 2021) to obtain the initial pairwise representations. At each Transformer layer, the pair representations are updated by adding the attention matrix calculated from cell representations to it. Then, the updated pair representations are used as the actual attention scores for updating cell representations. This approach of using pair representations as attention bias has been shown to be effective in various important works such as Graphformer (Ying et al., 2021) and Alphafold series (Jumper et al., 2021; Abramson et al., 2024). Moreover, given the strong locality of cell-cell interactions (Zormpas et al., 2023), we envision that the distance information is suitable input for the pair representation in handling ST data. More details about the SE(2) Transformer architecture can be found in Appendix A.2.

We train the SE(2) Transformer $f_{ST}$ to encode each sub-slice $S^{(i)}$, obtaining cell representations $Y_{cell}^{(i)}$ and pairwise representations $Y_{pair}^{(i)}$, as follows:

$$Y_{cell}^{(i)}, Y_{pair}^{(i)} = f_{ST}\left(F^{(i)}, P^{(i)}\right)$$

In pretraining, $Y_{cell}^{(i)}$ and $Y_{pair}^{(i)}$ are used to calculate the loss function. In downstream applications, we perform sub-slice division once and calculate the cell embeddings for each cell from the corresponding sub-slice.

### 3.3. Pretraining Objectives and Strategies

**Domain Adaptation Objectives**   During domain adaptation, we adopt the masked gene modeling objective in Geneformer (Theodoris et al., 2023), which is to predict the

randomly masked genes in the input gene sequence arranged by relative expression levels. In addition, inspired by Zhao et al., we introduce a self-supervised contrastive learning task (Gao et al., 2021) to enhance the representation quality of the cell encoder. See Appendix A.1 for details.

**Multi-scale ST Representation Learning Objectives**   In order to better capture multi-scale information from ST data, we need to design objectives that harvest supervision signals from both gene expressions and cell coordinates. To this end, we design the following two tasks, masked cell modeling (**MCM**) and pairwise distance recovery (**PDR**):

In Masked Cell Modeling (**MCM**), we randomly mask 10% of the gene expression embeddings $F^{(i)}$ in the sub-slice, and use SToFM's output representations $Y_{cell}^{(i)}$ to predict the masked embeddings through a regression head. Given the impracticality of using micro-scale information to reconstruct macro-scale information, we do not mask virtual cell embeddings. We use the Mean Squared Error (MSE) loss function. The objective is defined as:

$$\mathcal{L}_{MCM} = \frac{1}{\|\mathcal{M}_1\|} \sum_{j \in \mathcal{M}_1} \left(\|\hat{F}_j^{(i)} - F_j^{(i)}\|_2\right)^2$$

Where $\mathcal{M}_1$ is the set of masked cells, $F_j^{(i)}$ is the $j$-th cell embedding, and $\hat{F}_j^{(i)}$ is the restored $j$-th cell embedding predicted with the $j$-th cell representation from $Y_{cell}^{(i)}$.

In Pairwise Distance Recovery (**PDR**), we randomly select 10% cells in the sub-slice and add Gaussian noise to their 2D coordinates $P^{(i)}$, which modifies the corresponding rows and columns of the distance matrix $D^{(i)}$. Our model then attempts to reconstruct the unperturbed distance matrix through a regression head using the pair representation $Y_{pair}^{(i)}$. For the same cell, we make sure not to mask its embedding and perturb its coordinates at the same time. We use the MSE loss function. The objective is defined as:

$$\mathcal{L}_{PDR} = \frac{1}{\|\mathcal{M}_2\|} \sum_{(j,k) \in \mathcal{M}_2} \left(\|\hat{D}_{jk}^{(i)} - D_{jk}^{(i)}\|_2\right)^2$$

Where $\mathcal{M}_2$ is the set of perturbed elements in the distance matrix, $D_{jk}^{(i)}$ is the distance between cells $j$ and $k$ calculated from the unperturbed positions, and $\hat{D}_{jk}^{(i)}$ is the restored distance between cells $j$ and $k$, predicted by a regression head using the pair representations $Y_{pair}^{(i)}$.

**Pretraining Strategies**   We adopt some strategies to help stabilize the training and accelerate convergence. In the early stage of pretraining, since the domain-adapted cell encoder has undergone pretraining while the SE(2)-

---

[2]This architecture is originally an SE(3) Transformer, but it can be applied to 2D scenes with little architectural modifications.

Transformer is trained from scratch, we perform synchronization by freezing the cell encoder and directly using $F^{(i)}$ as the inputs of the SE(2) Transformer. Once the training of the SE(2) Transformer has mostly converged, we tune the parameters of both the cell encoder and the SE(2) Transformer. Considering the high computational cost of the cell encoder, we perform the second-time cell encoding mentioned in Sec. 3.2 only on some randomly sampled cells within the sub-slice. This allows us to update both the cell encoder and the SE(2) Transformer simultaneously via backpropagation, improving the model's learning capability.

# 4. Experiments

## 4.1. Experiment settings

**Construction of SToCorpus-88M**  We organize publicly available ST data from multiple sources (Xu et al., 2024; Yuan et al., 2023; Biology et al., 2023; 10xGENOMICS, 2025; Vizgen, 2025; nanoString, 2025; SeekGene, 2025). As some low-resolution ST data cannot accurately reflect gene expression at the gene level, we only select sequencing methods with single-cell resolution or those whose spot diameters are close to cell diameters. Specifically, we select six sequencing methods and construct SToCorpus-88M, the largest ST pretraining corpus, which contains approximately 2,000 tissue slices and 88 million cells, surpassing the largest ST corpus to date by 1.6 fold (Schaar et al., 2024). SToCorpus-88M includes slices from human and mouse, and gene expression profiles are aligned by homologous genes (Harrison et al., 2024). We standardize the corpus using gene names and Ensembl IDs, and filter out cells with less than 100 expressed genes to ensure data quality. Please refer to Appendix D.2 for more details.

**Pretraining Setup**  Pretraining is performed on SToCorpus-88M after removing the data used for downstream tasks. We perform domain adaptation for one epoch. Then, we perform multi-scale ST representation learning for three epochs, where the cell encoder is frozen in the first two epochs, following the strategy in Sec. 3.3. The pretraining is performed with 4 NVIDIA Tesla A100 GPUs and takes approximately 20 days. Detailed pretraining hyperparameters are provided in Appendix C.1.

**Baseline Selection and Downstream Task Setup**  For baseline comparisons, we select pretrained models that only process single-cell transcriptomics data or ST data, without incorporating other modalities like text (Zhao et al., 2024; Levine et al., 2023) or image (Lin et al., 2024). The chosen baselines are single-cell models Geneformer (Theodoris et al., 2023) and scGPT (Cui et al., 2023), the ST-specialized single-cell model Nicheformer (Schaar et al., 2024), and the first model combining gene expression and spatial information in ST data, CellPLM (Wen et al., 2023). These models have demonstrated superior performance over traditional methods across various tasks. Since these models differ in parameters and architecture, their full-parameter fine-tuning performance could be heavily influenced by hyperparameter settings and fine-tuning strategies. To ensure a fair comparison of representation quality, we perform head tuning on downstream tasks by training task-specific heads on top of the pretrained model outputs (Wei et al., 2021). As a supplement, we also conduct full-parameter fine-tuning experiments in Appendix B.1.

## 4.2. Tissue Region Semantic Segmentation

Similar to image or point cloud semantic segmentation, tissue region semantic segmentation is one of the key tasks in spatial transcriptomics (Zhou et al., 2024), aiming to determine which structural or functional region each cell belongs to. Accurate region segmentation not only helps reveal spatial transcriptional patterns within tissues but also provides a foundation for understanding the functional specialization of cells in biological processes. We evaluate SToFM's performance on tissue region semantic segmentation in two biologically significant experimental scenarios: human embryonic structure segmentation and DLPFC layer segmentation. For each task, we collect a dataset consisting of four slices and evaluate the model performance on each slice separately. Additionally, to assess the model's transferability, we employ a more challenging cross-slice setting, where the classification head is trained on three slices and tested on the remaining slice.

**Human Embryonic Structure Segmentation**  Human embryonic development is a highly dynamic and complex process. Accurately segmenting human embryonic structures aids in understanding cellular differentiation and tissue formation during development, and also uncovers the origins and regulatory mechanisms of key organs and tissues. In this experiment, we use human embryonic Stereo-seq ST data to perform structure segmentation (Pan et al., 2023; HESTA, 2025).

**DLPFC Layer Segmentation**  The dorsolateral prefrontal cortex (DLPFC), which is a key part of the brain's prefrontal cortex, plays a crucial role in various higher-order cognitive functions such as working memory and attention regulation. The DLPFC is part of the brain's neocortex and has a typical six-layer laminar structure. In this experiment, we use a human DLPFC 10x Visium ST dataset (Maynard et al., 2021), which has been manually annotated with layer information in previous studies. It is worth noting that 10x Visium data is not used during the pretraining process of SToFM or any of the baseline models. Therefore, this experiment also evaluates how well the models can adapt to data from a

*Table 1:* Performance of tissue region semantic segmentation on human embryo and DLPFC slices. Expr: gene expressions. Pos: cell positions. M-S: Multi-scale. Acc: accuracy. $F_1$: macro $F_1$ score.

| | **Embryonic Structure Segmentation** | | | | | | | | | | | | | | | | | |
|---|---|---|---|---|---|---|---|---|---|---|---|---|---|---|---|---|---|---|
| Model | ST Pretraining | | | Embryo1 | | Embryo2 | | Embryo3 | | Embryo4 | | **Average** | | **Cross-slice** | |
| | Expr | Pos | M-S | Acc | $F_1$ | Acc | $F_1$ | Acc | $F_1$ | Acc | $F_1$ | Acc | $F_1$ | Acc | $F_1$ |
| scGPT | ✗ | ✗ | ✗ | 0.7906 | 0.7872 | 0.7987 | 0.7210 | 0.7731 | 0.7146 | 0.8084 | 0.7572 | 0.7927 | 0.7450 | 0.5752 | 0.3947 |
| Geneformer | ✗ | ✗ | ✗ | 0.7891 | 0.7789 | 0.7810 | 0.6854 | 0.8028 | 0.7572 | 0.7986 | 0.7651 | 0.7929 | 0.7467 | 0.5293 | 0.3745 |
| Nicheformer | ✔ | ✗ | ✗ | 0.7260 | 0.7119 | 0.6870 | 0.5858 | 0.7180 | 0.6336 | 0.7306 | 0.6842 | 0.7154 | 0.6539 | 0.5359 | 0.3718 |
| CellPLM | ✔ | ✔ | ✗ | 0.8358 | 0.8261 | 0.8179 | 0.7325 | 0.8068 | 0.7566 | 0.8139 | 0.7737 | 0.8186 | 0.7722 | 0.5597 | 0.3985 |
| **SToFM** | ✔ | ✔ | ✔ | **0.8514** | **0.8312** | **0.8380** | **0.8014** | **0.8175** | **0.7716** | **0.8380** | **0.8143** | **0.8362** | **0.8046** | **0.6059** | **0.4588** |
| | **DLPFC Layer Segmentation** | | | | | | | | | | | | | | | | | |
| Model | ST Pretraining | | | DLPFC1 | | DLPFC2 | | DLPFC3 | | DLPFC4 | | **Average** | | **Cross-slice** | |
| | Expr | Pos | M-S | Acc | $F_1$ | Acc | $F_1$ | Acc | $F_1$ | Acc | $F_1$ | Acc | $F_1$ | Acc | $F_1$ |
| scGPT | ✗ | ✗ | ✗ | 0.6943 | 0.6599 | 0.6565 | 0.5908 | 0.6919 | 0.6151 | 0.6696 | 0.6055 | 0.6781 | 0.6178 | 0.6689 | 0.5885 |
| Geneformer | ✗ | ✗ | ✗ | 0.6404 | 0.5770 | 0.6286 | 0.5464 | 0.6373 | 0.5677 | 0.6157 | 0.5513 | 0.6305 | 0.5606 | 0.5981 | 0.5440 |
| Nicheformer | ✔ | ✗ | ✗ | 0.5712 | 0.5319 | 0.5942 | 0.5300 | 0.5588 | 0.5192 | 0.5648 | 0.5141 | 0.5723 | 0.5238 | 0.5809 | 0.5138 |
| CellPLM | ✔ | ✔ | ✗ | 0.7005 | 0.6553 | 0.6960 | 0.5923 | 0.7143 | 0.6482 | 0.6725 | 0.5918 | 0.6958 | 0.6219 | 0.6908 | 0.5953 |
| **SToFM** | ✔ | ✔ | ✔ | **0.7082** | **0.6755** | **0.6974** | **0.6453** | **0.7157** | **0.6659** | **0.6856** | **0.6274** | **0.7014** | **0.6535** | **0.6981** | **0.6437** |

completely new sequencing technique.

We use classification accuracy and $F_1$ score as evaluation metrics. The results in Table 1 demonstrate that SToFM outperforms existing methods across different tissue region semantic segmentation tasks. As the only baseline that uses both gene expression and location information of the cells, CellPLM achieves second-best performance in most tasks, which further demonstrates the importance of jointly incorporating these two types of information. Notably, we observe that SToFM surpasses other models to a greater extent in the cross-slice setting, demonstrating its robustness and transferability. We attribute this to the integration of micro- and macro-scale information in SToFM, since tissue morphology and cell-cell interaction patterns are more likely to transfer across different ST slices. In conclusion, SToFM achieves high-precision tissue region segmentation in various complex biological contexts, providing a powerful tool for the biological interpretation of ST data.

### 4.3. Cell Type Annotation in Spatial Transcriptomics

Cell type annotation is a fundamental problem in ST research. Compared to scRNA-seq data, the gene expression profiles of ST data have lower data quality due to fewer captured genes or a higher probability of dropout zeros. This poses challenges for cell type annotation based on gene expression profiles. For this task, we use two mouse brain datasets obtained using the Stereo-seq and SeekSpace technologies (Cheng et al., 2022; SeekGene, 2025). These two datasets have been manually annotated as 25 and 8 different cell types, respectively, as ground truth. The results in Table 2 show that our model significantly outperforms existing methods on both datasets. This confirms that incorporating spatial information can help improve cell type annotation on ST data. Specifically, the tissue region where the cells are located and the cell type composition in the cell's neighborhood can aid in inferring cell types even in the scenes of

*Table 2:* Performance of cell type annotation on mouse brain slices.

| **Models** | Brain1 | | Brain2 | |
|---|---|---|---|---|
| | Accuracy | Macro $F_1$ | Accuracy | Macro $F_1$ |
| Geneformer | 0.5646 | 0.3853 | 0.8540 | 0.6742 |
| Nicheformer | 0.5186 | 0.3550 | 0.8447 | 0.7575 |
| CellPLM | 0.6001 | 0.4186 | 0.9256 | 0.7332 |
| **SToFM** | **0.6349** | **0.4951** | **0.9289** | **0.8362** |

low-quality gene expression data.

### 4.4. Zero-shot Clustering and Visualization

Clustering and visualizing cells is important in bioinformatics analysis. In this section, we use cell representations from SToFM and baseline models for clustering and visualization, and evaluate how well the clustering results align with the ground truth of cell types on a MERFISH mouse brain slice (Allen et al., 2023). The experimental results in Fig. 3 demonstrate that clustering based on SToFM cell embeddings achieves the best performance in terms of NMI and ARI scores when compared to true cell type labels. The visualization further illustrates the ability of SToFM to produce high-quality cell embeddings, where each cell type forms a distinct and tight cluster in the SToFM embedding space. Interestingly, the cell representations of SToFM also reflect some relationships between cells. For example, *oligodendrocyte progenitor cell* shares a common progenitor with *neuron* and can further differentiate into *oligodendrocyte*.

### 4.5. Spatial Deconvolution

As described in Appendix D.1, in some ST methods, each data point actually represents a small region called a spot which contains parts of different cells. Therefore, a spot can be described as a proportion of various cell types. Predicting the cell type proportion in ST data is known as spatial deconvolution. Considering that foundation models can

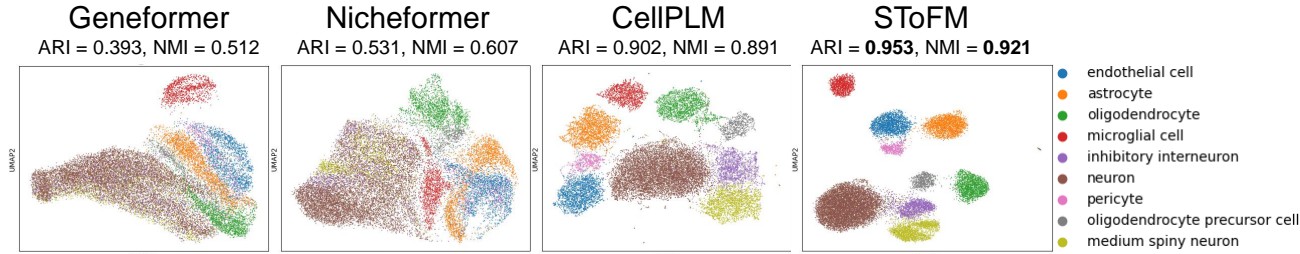

*Figure 3:* UMAP visualization of cell representations on a mouse brain slice, colored by cell types. ARI and NMI scores were calculated with Leiden clustering results and cell type labels.

*Table 3:* Performance of spatial deconvolution on mouse liver slices.

| Models | LiverSample | | | LiverCross | | |
|---|---|---|---|---|---|---|
| | Pearson↑ | RMSE↓ | MAE↓ | Pearson↑ | RMSE↓ | MAE↓ |
| Geneformer | 0.7152 | 0.0909 | 0.0446 | 0.6906 | 0.0970 | 0.0456 |
| Nicheformer | 0.7143 | 0.0909 | 0.0440 | 0.6938 | 0.0976 | 0.0447 |
| CellPLM | 0.7458 | 0.0865 | 0.0423 | 0.7248 | 0.0921 | 0.0441 |
| SToFM | **0.7666** | **0.0833** | **0.0408** | **0.7546** | **0.0867** | **0.0430** |

*Table 4:* Performance of imputation on human skin slices.

| Models | SkinCross | | |
|---|---|---|---|
| | Pearson↑ | RMSE↓ | MAE↓ |
| Geneformer | 0.3610 | 0.3671 | 0.1029 |
| Nicheformer | 0.2829 | 0.3772 | 0.1163 |
| CellPLM | 0.3985 | 0.3758 | 0.0905 |
| SToFM | **0.4877** | **0.3356** | **0.0886** |

*Table 5:* Ablation study of the effect of different scales on SToFM performance. VCs: virtual cells. DA: domain adaptation.

| Ablated Models | Multi-scale | | | EmbryoCross | Brain1 |
|---|---|---|---|---|---|
| | Gene | Micro | Macro | Macro $F_1$ | Macro $F_1$ |
| Cell encoder w/o DA | ✗ | ✗ | ✗ | 0.3745 | 0.3853 |
| Cell encoder w/ DA | ✔ | ✗ | ✗ | 0.4155 | 0.4725 |
| SToFM w/o VCs | ✔ | ✔ | ✗ | 0.4291 | 0.4893 |
| **SToFM** | ✔ | ✔ | ✔ | **0.4588** | **0.4951** |

generate consistent representations for data from different slices, using foundation models to transfer deconvolution results from labeled slices to unlabeled slices provides a promising solution for this task. This section explores the potential of using SToFM for ST deconvolution in a simple quantitative experiment. The experiment is conducted on a Stereo-seq mouse liver dataset (Wu et al., 2024), testing the models' ability to transfer deconvolution labels both within the same slice and across different slices. The ground truth is annotated by biologists (Wu et al., 2024). We adopt 3 evaluation metrics for regression: Pearson correlation, RMSE and MAE. Table 3 reports the results, showing that SToFM achieves the best performance in both within-slice and cross-slice deconvolution tasks. In addition, SToFM achieves more improvement under the cross-slice setting, which further emphasizes its excellent transferability.

### 4.6. Spatial Transcriptomics Imputation

ST technologies often cannot capture a large number of genes while maintaining high resolution, posing challenges for data analysis. ST imputation, i.e., inferring the uncaptured gene expression levels, is an important task in the analysis of these data (Abdelaal et al., 2020; Jiang et al., 2024). Similar to 4.5, we explore the potential of SToFM for ST imputation. We collect two 10x Xenium human skin datasets (10xGENOMICS, 2025) to be used as a training dataset and a test dataset. Each cell in these datasets has 377 known gene expressions, of which we randomly select 50

as labels and the remaining 327 as inputs. The evaluation metrics are Pearson correlation, RMSE and MAE. The experimental results in Table 4 show the excellent performance of SToFM on this task.

### 4.7. Ablation Study

In order to demonstrate the performance improvement resulting from modeling information at multiple scales, we conduct a set of ablation experiments as shown in Table 5. Specifically, we first ablate the macro-scale information by removing the virtual cells. Then, we ablate the model's ability to jointly model multiple cells at the micro scale by removing the SE(2) Transformer. Finally, we remove domain adaptation to test the model's original ability to handle gene-scale expression profiles. The ablation experiments are performed on two representative tasks: embryo structure segmentation and cell type annotation.

The results have demonstrated the performance gains brought by each scale, confirming that the integration of multi-scale information is helpful for various downstream tasks. In addition, the cross-slice results also prove that more macroscopic information greatly enhances the transferability of the model.

## 5. Conclusions and Limitations

In this work, we propose SToFM, a multi-scale ST foundational model, pretrained on our large-scale ST corpus, SToCorpus-88M. By effectively capturing and integrating information across different scales, SToFM achieves a comprehensive understanding of biological information in ST data. It demonstrates outstanding performance in various downstream tasks, introducing a new perspective for the analysis of ST data and driving progress in the field.

Currently, SToFM still has some limitations. SToFM only

considers three scales: macro, micro, and gene. In the future, we may consider modeling and integrating more scales using methods like image pyramid (Adelson et al., 1984), or introduce causal machine learning methods to model gene regulatory relationships (Tejada-Lapuerta et al., 2025; Zhong et al., 2024). Furthermore, it is also possible to enhance the model by using prior biological knowledge or other modalities, such as known ligand-receptor pairs or pathological images. We reserve these explorations for future works.

## Code Availability

SToFM will soon be added to the OpenBioMed toolkit: https://github.com/PharMolix/OpenBioMed.

Code is available at: https://github.com/PharMolix/SToFM.

## Acknowledgements

This research is supported by the National Key R&D Program of China (No. 2022YFF1203002) and PharMolix Inc.

## Impact Statement

This paper presents work whose goal is to advance the field of Machine Learning. There are many potential societal consequences of our work, none which we feel must be specifically highlighted here.

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

# Appendix

# A. Architecture of the Model Components

In this section, we describe the detailed architecture of the model components.

### A.1. Cell Encoder

We use the single-cell foundation model, Geneformer (Theodoris et al., 2023), as the cell encoder of SToFM. The encoding process of Geneformer is as follows: **(1)** Normalize the gene expression matrix for each cell. Then, divide the expression level of each gene by the median of its non-zero expression values across all cells to obtain the relative expression level of each gene in the cell. **(2)** Rank the genes in each cell based on their relative expression levels to generate a sequence representation for the cell. **(3)** Use a BERT model to encode the serialized cell representation.

In the domain adaptation process, we follow Geneformer's pretraining task of Masked Gene Modeling(MGM), where some genes in the cell sequence representation are masked, and the model predicts the masked genes to reconstruct them. Additionally, we incorporate the self-supervised Contrastive Learning (CL) approach (Gao et al., 2021; Zhao et al., 2024) to enhance the model's ability to capture global information. The loss function of domain adaptation ($\mathcal{L}_{\mathrm{DA}}$) is as follows:

$$\mathcal{L}_{\mathrm{DA}} = \mathcal{L}_{\mathrm{MGM}} + \mathcal{L}_{\mathrm{CL}} = \frac{1}{N}\sum_{i=1}^{N} H(v_{ij}, \hat{v}_{ij}) - \frac{1}{T}\sum_{i=1}^{T} \log \frac{\mathrm{e}^{\mathrm{sim}(F_i, F_i^+)/\tau}}{\sum_{j=1}^{T} \mathrm{e}^{\mathrm{sim}(F_i, F_j^+)/\tau}}$$

where $N$ is the number of masked genes, $v_{ij}$ and $\hat{v}_{ij}$ respectively represent the label and predicted probability of the $i$-th masked position being identified as the $j$-th gene, $H$ is the cross entropy loss function. $T$ is the batch size, $\mathrm{sim}$ is the cosine similarity function, $\tau$ is the temperature parameter, $F_i$ and $F_i^+$ represent the embedding of the $i$-th cell and its positive sample, respectively.

### A.2. SE(2) Transformer

We draw inspiration from the Uni-Mol framework (Zhou et al., 2023) to implement our SE(2) Transformer, using a simple yet effective architecture. The model structure is illustrated in Fig. A.1 and can be described as follows: **(1).** A distance matrix is used as the spatial input to ensure SE(2) invariance in a straightforward manner. **(2).** The initial cell input is obtained by adding the cell features to a learnable type embedding that distinguishes virtual cells from real cells. The distance matrix is passed through a learnable Gaussian module (Shuaibi et al., 2021; Zhou et al., 2023) to project it to the dimensionality of the model's attention heads, serving as the initial pair representation. **(3).** Both cell representations and pair representations are propagated between layers. In each layer, a self-attention matrix is computed for the cell representations and is added with the pair representation which serve as attention bias. The resulting combination is used to update both the pair representation and the attention matrix. This process is illustrated as follows:

$$R_{ij}^{l+1,h} = R_{ij}^{l,h} + \frac{Q_i^{l,h}(K_j^{l,h})^T}{\sqrt{d}},$$

$$\mathrm{Attention}_{ij}^{l,h} = \mathrm{softmax}(R_{ij}^{l+1,h})V_j^{l,h},$$

where $R^{l,h}$ is the pair representation matrix corresponding to the attention head $h$ in layer $l$, and $Q^{l,h}$, $K^{l,h}$, $V^{l,h}$ are the attention parameters corresponding to the attention head in layer $l$ and head $h$.

# B. More Experimental Results

### B.1. Full-parameter tuning

In the main text, we aim to fairly compare the representation learning abilities of SToFM with a basic model serving as the baseline. Therefore, we only perform head-tuning on downstream tasks. However, achieving better results in downstream tasks through full-parameter tuning is also an important use case for foundation models. Therefore, in this section, we evaluate the performance of full-parameter tuned SToFM on the cell type annotation task.

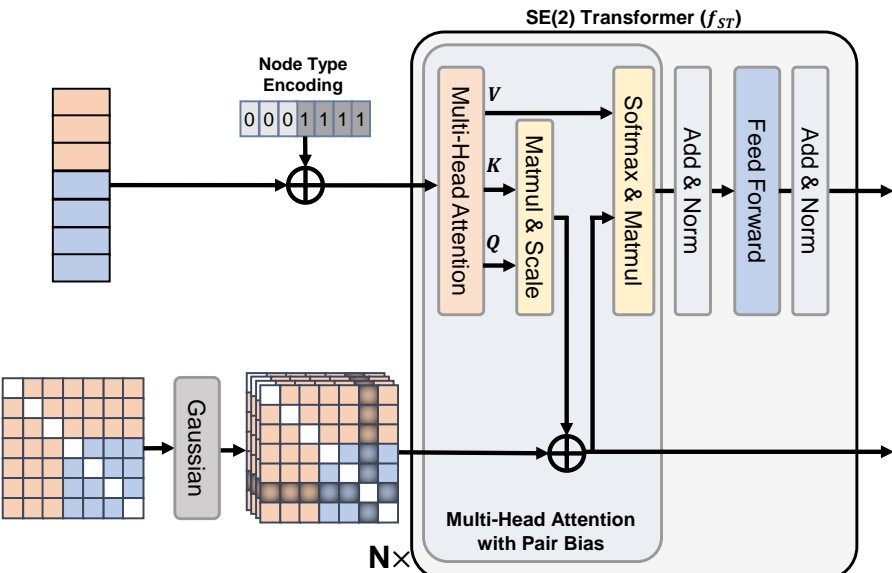

*Figure A.1:* The architecture of SE(2) Transformer in SToFM

*Table A.1:* Performance of cell type annotation on head tuning and full-parameter tuning.

| Models | Brain1 | | Brain2 | |
|---|---|---|---|---|
| | Accuracy | Macro $F_1$ | Accuracy | Macro $F_1$ |
| SToFM head tuning | 0.6349 | 0.4951 | 0.9289 | 0.8362 |
| SToFM full-parameter tuning | 0.6987 | 0.5532 | 0.9725 | 0.8903 |

## B.2. More Ablation Studies

**The resolution of clustering** and **the scale of sub-slices**. They are critical hyperparameters in SToFM. In our experiments, we used the Leiden algorithm for clustering, with the resolution set to 1.0, typically resulting in dozens of clusters. We ensured that each sub-slice contained approximately 1,000 cells, which aligns with the empirically suitable data scale for the SE(2) Transformer and includes sufficient microscopic information. Ablation experiments in Table A.2 confirmed that these hyperparameter settings fall within a reasonable range. Additionally, SToFM demonstrates robustness to changes in these hyperparameters within a certain range.

**Micro-scale components** and **spatial distance matrix**. We ablate the micro scale by limiting the scale of sub-slices to 1, to ensure that the cells could only interact with the virtual cells that represent macro-scale information, but not with other cells in the micro-environment. Experimental results are shown in Table A.3. A significant decrease in performance is observed, which demonstrates the effectiveness of incorporating the micro-scale information. Then, we conduct an ablation study on the spatial distance matrix. The spatial information is very underutilized in this case, only used to construct virtual cells and divide sub-slices. As shown in Table A.3, the results demonstrate that removing the spatial distance matrix significantly decreases model performance.

**Sample rate of the second-time cell encoding**. The purpose of the second time cell encoding is to enable $\mathcal{L}_{MCM}$ and $\mathcal{L}_{PDR}$ to optimize the cell encoder through backpropagation. To balance the training cost and model performance, we use only a small number of cells for this computation, which is similar to selecting a smaller batch size to update the cell encoder. For the selection of the sampling number, we determined the number of samples to be 12, given that the original Geneformer paper gave a training batch size of 12. Considering the computational cost, we test the impact of the sample number on a small amount of data (1/8 of the SToCorpus-88M), as shown in Table A.4. The results show that the model has some robustness to this hyperparameter, just as the batch size often only affects the convergence speed rather than the model performance. However, setting the sample size to 0, i.e. freezing the cell encoder, will lead to a decrease in model performance.

*Table A.2:* Ablation study of sub-slice scale and Leiden resolution.

| Sub-slice Scale | Leiden Resolution | Embryo2 | | EmbryoCross | |
|---|---|---|---|---|---|
| | | Accuracy | Macro $F_1$ | Accuracy | Macro $F_1$ |
| 10 | **1.0** | 0.7554 | 0.7066 | 0.5740 | 0.4291 |
| 100 | **1.0** | 0.8282 | 0.7418 | 0.5818 | 0.4510 |
| 500 | **1.0** | 0.8293 | 0.7899 | 0.5977 | 0.4475 |
| 2000 | **1.0** | 0.8018 | 0.7457 | 0.6002 | 0.4554 |
| **1000** | 0.1 | 0.8171 | 0.7755 | 0.5966 | 0.4410 |
| **1000** | 0.5 | 0.8014 | 0.7614 | 0.6008 | 0.4572 |
| **1000** | 1.5 | 0.8352 | 0.7523 | 0.6055 | 0.4403 |
| **1000** | **1.0** | **0.8380** | **0.8014** | **0.6059** | **0.4588** |

*Table A.3:* Ablation study of micro-scale components and spatial distance matrix.

| Model | Embryo2 Macro $F_1$ | EmbryoCross Macro $F_1$ |
|---|---|---|
| w/o micro | 0.721 | 0.425 |
| w/o spatial matrix | 0.721 | 0.413 |
| **SToFM** | **0.801** | **0.459** |

**Data volume**. We pretrain the multi-scale ST representation learning phase using 12.5% and 50% of the data, as shown in Table A.5. The results show that reduction in data volume led to a significant decrease in model performance. Considering that the SToCorpus-88M consists of approximately 2,000 ST slices, we believe that reducing the amount of data may reduce data diversity and limit the model's transferability.

$\alpha$ **in Algorithm 1**. We have conducted experiments to show how different $\alpha$ values affect the model's performance, as shown in Table A.6. The model has a certain robustness in alpha, and we believe this may be because cells that are closer in location are more likely to have similar gene expressions (Zormpas et al., 2023). The alpha=0.8 that we chose is essentially the optimal setting.

**Combination ratio of the two loss functions**. These two loss are relatively close in scale, and in our experiments, combining them in a 1:1 ratio allows both to converge normally. Considering the computational cost, we test the impact of the ratio of the two losses on the speed of convergence and the performance of the model on a small amount of data (1/8 of the SToCorpus-88M), as shown in Table A.7. ($\gamma$ is the loss ratio in $\mathcal{L} = \gamma \times \mathcal{L}_{MCM} + (1 - \gamma) \times \mathcal{L}_{PDR}$)

### B.3. Zero-shot clustering

Further than in Sec. 4.4, we visualize the model's representation of mixed data from two mouse brain slices in Fig. A.2.

It is worth noting that this is not a common task of batch integration in single-cell analysis. Our goal is not to make cells from different slices indistinguishable. Instead, we believe that the cell representation generated by SToFM contains some global information from the slices. As a result, as shown in the figure, SToFM establishes similar representations for cells of the same type from different slices, and preserve subtle differences brought by global slice information.

## C. Experiment Settings for Pretraining and Downstream Tasks

### C.1. Pretraining settings

The training process is conducted using the PyTorch framework. We utilize the AdamW optimizer, with a learning rate strategy that involved a warm-up phase followed by linear decay. Pretraining is carried out on four NVIDIA Tesla A100 GPUs, with both Domain Adaptation and pretraining taking approximately 10 days each to complete. Additional experimental configurations are detailed in Table A.8.

*Table A.4:* Ablation study of sample number of the second-time cell encoding (on 1/8 SToCorpus-88M).

| Sample number | Embryo2 Macro $F_1$ | EmbryoCross Macro $F_1$ |
|:---:|:---:|:---:|
| 0 | 0.722 | 0.417 |
| 4 | 0.754 | 0.424 |
| 12 | 0.758 | 0.423 |

*Table A.5:* **Ablation study of data volume.** We use 12.5% and 50% of the SToCorpus-88M for multi-scale ST representation learning pre-training, and compare them with model pre-trained on the full dataset. The results show that a larger pre-training dataset can improve model performance. We attribute this to the **increased diversity** of the data.

| Data volume | Embryo2 Macro $F_1$ | EmbryoCross Macro $F_1$ |
|:---:|:---:|:---:|
| 12.5% | 0.758 | 0.423 |
| 50% | 0.782 | 0.450 |
| 100% | 0.801 | 0.459 |

## C.2. Downstream tasks settings

For downstream tasks, we adhere to the following standardized settings:

- All datasets are filtered to remove special categories such as "Other" or "Unknown" and categories with less than 1% of the total samples.

- Tasks with inherent randomness are run three times, and the results are averaged.

- For tasks involving splitting the same dataset into training and testing sets, an 8:2 random split is applied. Additionally, 10% of the training set is randomly selected as a validation set, and an early stopping strategy is applied based on performance on the validation set.

- Both the classification and regression heads are implemented as three-layer fully connected neural networks with LeakyReLU activation functions.

## C.3. Details of multi-class classification tasks

Tissue region semantic segmentation (Sec. 4.2) and cell type annotation (Sec. 4.3) are both multi-class classification tasks. The number of categories are shown as Table A.9. We filtered out categories with fewer cells than 1% of the total number of cells.

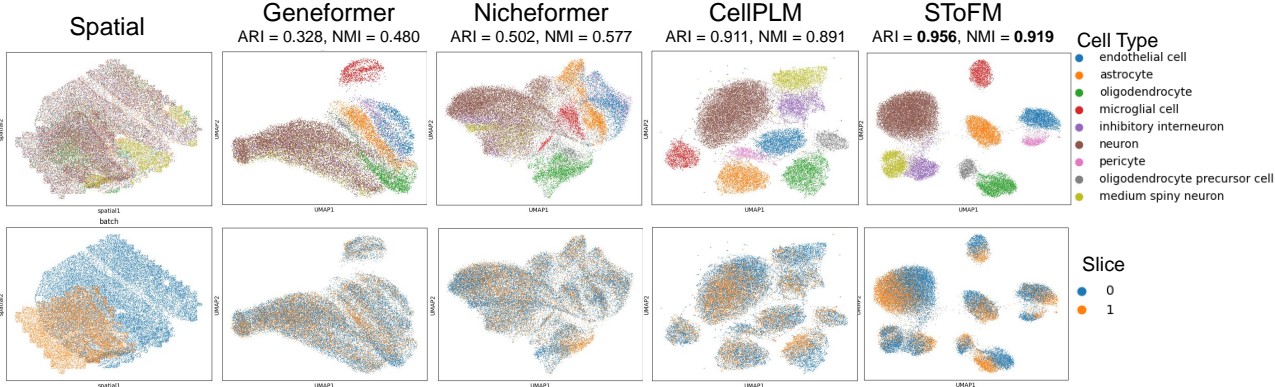

*Figure A.2:* UMAP visualization of cell representations on two mouse brain slices, colored by cell types and slices.

*Table A.6:* Ablation study of $\alpha$ in Algorithm 1.

| $\alpha$ | Embryo2 Macro $F_1$ | EmbryoCross Macro $F_1$ |
|---|---|---|
| 0 | 0.751 | 0.435 |
| 0.2 | 0.767 | 0.458 |
| 0.4 | 0.778 | 0.440 |
| 0.6 | 0.796 | 0.436 |
| 0.8 | 0.801 | 0.459 |
| 1.0 | 0.769 | 0.448 |

*Table A.7:* Ablation study of $\alpha$ in Algorithm 1.

| $\gamma$ | Embryo2 Macro $F_1$ | EmbryoCross Macro $F_1$ |
|---|---|---|
| 0 | 0.683 | 0.409 |
| 0.2 | 0.729 | 0.423 |
| 0.5 | 0.758 | 0.423 |
| 0.8 | 0.753 | 0.429 |
| 1.0 | 0.701 | 0.415 |

## D. Datasets

### D.1. Sequencing techniques and data characteristics in ST

Spatial transcriptomics is an advanced technology that allows simultaneous acquisition of spatial tissue information and gene expression data. The main sequencing technologies can be categorized into image-based single-cell resolution techniques and sequencing-based Spot techniques (Li et al., 2023).

**Image-based single-cell resolution technologies** These technologies typically rely on fluorescence in situ hybridization (FISH) techniques, such as MERFISH (Chen et al., 2015), 10x Xenium and CosMx (He et al., 2022). By labeling specific RNA molecules and imaging them, these technologies achieve spatial expression profiling at single-cell resolution. Their advantage lies in their high resolution, enabling precise localization of RNA within individual cells, but the number of genes that can be analyzed is usually limited. Some recent studies have compiled some image-based ST data (Jaume et al., 2024; Chen et al., 2024).

**Sequencing-based spot technologies** Sequencing-based technologies typically involve placing capture areas of a certain size, called spots, on tissue sections (e.g., 10x Visium (Ståhl et al., 2016)). Each spot covers multiple cells, with a size of approximately 50-100 microns. Newer techniques like Slide-seq (Stickels et al., 2021) use smaller beads with a diameter of about 10 microns, significantly improving resolution. Additionally, Stereo-seq (Chen et al., 2022) leverages nanometer-scale lithography arrays, achieving spot sizes of 220 nanometers, approaching single-cell or even subcellular resolution. SeekSpace uses a special positional probe method to achieve a single-cell resolution sequencing-based method.

Considering that some low-resolution technologies have spot diameters much larger than cell diameters, which cannot accurately reflect the gene expression distribution at the single-cell level, we exclude these data when constructing the datasets for pretraining and the vast majority of downstream tasks. Specifically, we only use low-resolution data in the DLPFC layer segmentation task (Sec. 4.2) to evaluate the model's transferability on this type of data.

### D.2. SToCorpus-88M

We selected data obtained from 6 high-resolution sequencing technologies to construct SToCorpus-88M. These include image-based technologies MERFISH, 10x Xenium, and CosMx, as well as sequencing-based technologies Stereo-seq, Slide-seqv2, and SeekSpace. Among them, MERFISH, 10x Xenium, CosMx, and SeekSpace are single-cell resolution data, while Stereo-seq and Slide-seqv2 are data where the diameter of spots is close to the diameter of cells.

Our data sources include six public databases or data release projects. We also obtain some unpublished SeekSpace data. The

*Table A.8:* Experiment Configurations

| | **Hyperparameter** | **Value** |
|---|---|---|
| | Leiden resolution | 1.0 |
| Multi-scale processing | $\alpha$ in Algorithm 1 | 0.8 |
| | Split scale | 1000 |
| | Vocab size | 25426 |
| | Hidden size | 512 |
| | Number of hidden layers | 12 |
| Cell Encoder | Max sequence length | 2048 |
| | Number of attention heads | 8 |
| | Dropout | 0.02 |
| | Hidden act | ReLU |
| | LayerNorm eps | 1e-12 |
| | Node Types | [Cell], [Virtual Cell], [PAD] |
| | Hidden size | 256 |
| | Number of hidden layers | 4 |
| SE(2) Transformer | Max sequence length | 2048 |
| | Number of attention heads | 8 |
| | Dropout | 0.1 |
| | Hidden act | GELU |
| | LayerNorm eps | 1e-5 |
| | Optimizer | AdamW |
| | Scheduler | Linear |
| Pretraining | Max learning rate | 1e-4 |
| | Warm up steps | 500 |
| | Batch size | 16 |
| | Gradient accumulation | 4 |

*Table A.9:* **Details of multi-class classification tasks.**

| Task | Data | Number of categories | Number of categories after filtering |
|---|---|---|---|
| Tissue region semantic segmentation (Sec. 4.2) | Embryo | 18 | 16 |
| Tissue region semantic segmentation (Sec. 4.2) | DLPFC | 6 | 6 |
| Cell type annotation (Sec. 4.3) | Brain1 | 25 | 21 |
| Cell type annotation (Sec. 4.3) | Brain2 | 8 | 6 |

data volume and corresponding sequencing technologies from each source are shown in Table A.10. We have removed the duplicated data obtained from different sources. In addition, individual slices in SToCorpus-88M are used for downstream tasks, and we exclude these data during pretraining.

These data are obtained from the following sources:

- 10xGENOMICS (10xGENOMICS, 2025): `10xgenomics.com/datasets`

- CELLxGENE (Biology et al., 2023): `cellxgene.cziscience.com`

- nanoString (nanoString, 2025): `nanostring.com/products/cosmx-spatial-molecular-imager/ffpe-dataset/`

- SODB (Yuan et al., 2023): `gene.ai.tencent.com/SpatialOmics/`

- STOmicsDB (Xu et al., 2024): `db.cngb.org/stomics/`

- Vizgen (Vizgen, 2025): `vizgen.com/data-release-program/`

- SeekSpace (SeekGene, 2025): not yet public, a demo can be found at `seekgene.com/sjzszx` and downloaded at `seekonetools-release.oss-cn-beijing.aliyuncs.com/demo_data/link/seekspace_WTH1092/WTH1092_filtered_feature_bc_matrix.zip`

*Table A.10:* SToCorpus-88M data sources and statistics.

| Sources | Sequencing technologies | Number of slices | Number of cells/spots |
|---|---|---|---|
| 10xGENOMICS | 10x Xenium | 20 | 6,496,946 |
| CELLxGENE | MERFISH, Slide-seqv2 | 762 | 23,531,670 |
| nanoString | CosMx | 6 | 1,118,500 |
| SODB | Slide-seqv2, Stereo-seq, MERFISH | 817 | 19,582,014 |
| STOmicsDB | Slide-seqv2, Stereo-seq, MERFISH | 242 | 24,768,293 |
| Vizgen | MERFISH | 32 | 12,344,989 |
| SeekSpace | SeekSpace | 33 | 338,953 |
| Total | 6 technologies | 1912 | 88,181,365 |

## D.3. Details of DownStream Task Datasets

**Embryonic Structure Segmentation**    In this experiment, we use a human embryo Stereo-seq ST dataset (Pan et al., 2023; HESTA, 2025), which can be downloaded at `db.cngb.org/stomics/hesta/`. The dataset includes human embryos at Carnegie Stages (CS) 12-23. Considering that the morphology of later embryos is too complex, we only use samples from CS12-13. We use four middle slices from CS12-13E2 (the slice position can be viewed from the visual function of the data website), i.e., CS12-13E2S3, CS12-13E2S4, CS12-13E2S5 and CS14-15E2S6, as Embryo1, Embryo2, Embryo3 and Embryo4 in Sec. 4.2. In addition, for the cross-slice task EmbryoCross in Sec. 4.2, we train on CS12-13E2S3, CS12-13E2S5, CS12-13E2S6, and test on CS12-13E2S4.

**DLPFC Layer Segmentation**    In this experiment, we use a human DLPFC 10x Visium ST dataset (Maynard et al., 2021). We select the four slices with the highest ids, 151673, 151674, 151675 and 151676, as DLPFC1, DLPFC2, DLPFC3 and DLPFC4 in section 4.2. For the cross-slice task DLPFCCross in Sec. 4.2, we train on slices 151674, 151675, and 151676, and test on slice 151673.

**Cell Type Annotation**    In this experiment, we use a mouse brain Stereo-seq ST slice (Cheng et al., 2022) and a mouse brain SeekSpace ST slice. The former can be downloaded from STOmicsDB (`db.cngb.org/stomics/datasets/ STDS0000139`). The latter has not been publicly released before and will be released along with this paper. Both slices have been processed to single-cell resolution.

**Zero-shot Clustering**    In this experiment, we use a mouse brain MERFISH ST dataset (Allen et al., 2023), which can be downloaded from SODB (`gene.ai.tencent.com/SpatialOmics/dataset?datasetID=184`). In Sec. 4.4, we use the first slice of this dataset, MsBrainAgingSpatialDonor_1_0. In Sec. B.3, we use two slices with some differences, MsBrainAgingSpatialDonor_1_0 and MsBrainAgingSpatialDonor_4_0.

**Spatial Deconvolution**    In this experiment, we use a mouse liver Stereo-seq ST dataset (Wu et al., 2024), which can be downloaded from STOmicsDB (`db.cngb.org/stomics/datasets/STDS0000239`). We used the scores corresponding to the following 17 cell types as regression labels: *Cholangiocyte*, *Monocyte*, *KC*, *Fibroblast*, *HSC*, *LSEC*, *B_cell*, *T_cell*, *LPLC*, *PC_LVEC*, *PP_LVEC*, *Neutrophil*, *NK*, *cDC*, *pDC*, *ILC1*, *VSMC*. We select the slice D17_FR4 as LiverSample in Sec. 4.5. For the cross-slice task LiverCross in Sec. 4.5, we train on slice D0_DY1 and test on slice D0_DY2.

**Imputation**    In this experiment, we use two human skin 10x Xenium ST slices (10xGE-NOMICS, 2025), which can be downloaded from `www.10xgenomics.com/datasets/ human-skin-data-xenium-human-multi-tissue-and-cancer-panel-1-standard`. For the cross-slice task SkinCross in Sec. 4.6, we train on slice Sample2, and test on slice Sample1.

