# OpenReview forum: "SToFM: a Multi-scale Foundation Model for Spatial Transcriptomics"
_ICML.cc/2025/Conference — ICML 2025 poster_

### Official Review · Reviewer_tnYZ · 2025-03-10

**Overall Recommendation:** 4

**Summary:**

The authors introduce SToFM, a single-cell foundation model that incorporates not only the single-cell expression data but also their spatial locations. They propose SToCorpus-88M, one of the largest single-cell pretraining datasets curated to date, and also pretrain their model on the large pretraining dataset. It is demonstrated that SToFM is able to outperform a lot of recent SOTA ST FM such as Geneformer, Nicheformer, and scGPT.

**Claims And Evidence:**

Most are well-supported - However, I think the claim that SToCorpus-88M is the largest high-resolution ST pretraining corpus to date needs to be thoroughly validated.

**Essential References Not Discussed:**

I think maybe recently-published public ST benchmarks (although they are predominantly Visium-based) deserve some references.

[1] Jaume, Guillaume, et al. "Hest-1k: A dataset for spatial transcriptomics and histology image analysis." Advances in Neural Information Processing Systems 37 (2024): 53798-53833.
[2] Chen, Jiawen, et al. "STimage-1K4M: A histopathology image-gene expression dataset for spatial transcriptomics." ArXiv (2024): arXiv-2406.

**Experimental Designs Or Analyses:**

I think the explanations on each experiment is quite lacking. For instance, for morphological segmentation, it is unclear how many regions are being segmented. For imputation, it's unclear how SToFM is being used to infer the ST for certain cells. In short, I have doubts that the readers can "reproduce" the experimental results just based on what is provided in the paper.

**Methods And Evaluation Criteria:**

Yes, the authors validate SToFM on diverse tasks such as ST imputation, morphological segmentation, and etc. and show universally better performance.

**Other Comments Or Suggestions:**

- How are the authors dealing with batch effect that arises from integrating 7 different sequencing platforms? There wasn't a reference to any efforts for batch effect correction, so I think it's very important.
- Continual training can easily lead to catastrophic forgetting - Have the authors observed this + What have authors done to prevent any form of collapse?
- scGPT in Table 1 says it doesn't use expression values - but scGPT uses expression values as well. What did the authors mean to say here?
- Leiden clustering cannot control for exact number of clusters, i.e., virtual cells in each sub-slice. How do authors control for this?
- Are the authors planning to release SToCorpus-88M publicly?

**Other Strengths And Weaknesses:**

While there are many strengths in this work, as demonstrated with diverse evaluation tasks against many SOTA baselines, there are several factors in this study that prevents me from giving higher score.

**Lack of clarity & details**: The technical details behind SToFM and the experiment details are quite lacking, to the point that it is a bit hard to understand and replicate what the authors have proposed. For instance, in domain-adaptation phase, there are at least 7 different platforms from which the data comes from - Is the domain-adaptation performed on all 88M points? In what sequence? Does each batch randomly sample from 88M point each time? Why only one epoch?

How are the sub-slice generated exactly? The authors only mention that it roughly contains 1,000 genes per sub-slice and it's probably hard for readers to replicate.

For the experiments, it's unclear what number of morphological classes the segmentation is done on - Is it binary, due to the use of F1 and acc? But DLPFC typically has at least 5~6 morphological classes. How are the ST imputation done? How are the 327 inputs used to impute for 50 outputs?

In short, I would cut down verbose explanations on the architecture and paradigm and focus on providing more technical/experimental details.

**Lack of ablations**: Right now, it feels like SToFM is a combination of existing parts (geneformer & SE(2) encoder & masking loss) put together. While I think that is fine and how the field advances, I feel like there is lack of motivation/explanation behind each of the component design choices. To really make strong claims about SToFM, I think the authors need to perform more extensive ablations (other than just cell/micro/macro ablations)
- Data efficiency: That 88M is very big size is good, but is this really what drives the performance? Especially if the authors are only performing one epoch of pretraining. I would like the authors to try pretraining SToFM with only fractions of data (e.g., 12.5%, 25%, 50%) to see whether SToFM really follows the scaling law and that all 88 M datapoints are needed.
- SE(2) Transformer: Can we try with different architecture (e.g., without spatial distance) to really show that the distance information is really required? I feel like expression reconstruction alone could be sufficient, but I might be wrong.
- Masking ratio: The masking ratio is universally 10%. Given that typical use of Masked Image Modeling use higher masking ratio than 10%, I would like the authors to show the effect of this.
- Geneformer initialization: What if it is randomly-initialized? What if other pretraining weights are used?

**Questions For Authors:**

Please see above

**Relation To Broader Scientific Literature:**

I think it is well-placed in terms of broad literature, especially with encouraging results against the SOTA baselines.

**Theoretical Claims:**

No theoretical claims are made.

---

> ### Author Rebuttal · Authors · 2025-03-31
>
> Dear Reviewer tnYZ:
>
> Thanks for your appreciation and detailed review! We try our best to response your questions. Due to space limitations, we include **tables and references in the anonymous link** https://anonymous.4open.science/api/repo/stofm-rebuttal/file/rebuttal-tnYZ.pdf?v=af96a9f5, and refer to them in the following as *Rebuttal-Table X* and [X], separately.
>
> >Q1: About SToCorpus-88M.
> - **Is it the largest?** To our best knowledge, it is the cell-resolution ST pretraining corpus that contains the largest number of both slices and data points.
> - **Will it be released?** Yes, it will be publicly released.
> - **Recently-published ST datasets.** Thank you! We will add citations to the revised manuscript.
>
> >Q2: Lack of clarity and details.
>
> Apologies for the lack of clarity. We promise to **add details to the revised version and release detailed code**.
> -  **Domain-adaptation**
> Domain adaptation performs an epoch in random order on all 88M data points. The loss had converged at the late stage of the first epoch.
> - **How are the sub-slice generated exactly?**
> First, we divide the slice into rectangles of the same size. Then, for each rectangle, we merge or split it according to the number of cells it contains. The code will be released for reproducing.
> -  **Number of morphological classes?**
> They are multi-classifacation tasks.  We have provided experimental details in the *Rebuttal-Table 1*.
> -  **How is the ST imputation done?**
> We trained a fully connected neural network with cell representations from SToFM to obtain 50 outputs, as a multi-target regression task.
>
> >Q3: Lack of ablations
>
> We will **add more ablation studies in the revised version**.
> -  **Data efficiency**
> Due to the high computational cost, we pretrain the multi-scale ST representation learning phase using 12.5% and 50% of the data during rebuttal, as shown in *Rebuttal-Table 2*. The results show that reduction in data volume led to a significant decrease in model performance. Considering that the SToCorpus-88M consists of approximately 2,000 ST slices, we believe that reducing the amount of data may reduce data diversity and limit the model's transferability. Additionally, we would like to clarify that, as mentioned in Section 4.1, we conducted 1 epoch and 3 epochs in each of the two training stages.
> -  **SE(2) Transformer**
> We conduct ablation studies on the PDR loss and the spatial distance matrix, as shown in *Rebuttal-Table 3*. For more analysis, please refer to the response Q1 to Reviewer LBxH. In addition, the effectiveness and efficiency of the SE(2) Transformer are discussed in detail in Uni-Mol[1]. And when incorporating spatial information in ST data, we primarily focus on the interactions between neighboring cells. Therefore, using the distance-based architecture is in line with the data characteristics.
> -  **Masking ratio**
> Masking ratios are set according to the specific data.  Each point in ST data contains high-dimensional features, which increase the data complexity. We had attempted to mask 20% of the cell expressions and perturb another 20% of the cell positions, but found training difficult to converge. In addition, [2] has demonstrated that models are usually robust to mask probabilities when they can converge properly.
> -  **Geneformer initialization**
> If randomly initialized,  it will incur expensive computational overhead to train a cell encoder from scratch. And due to the lower quality of gene expressions in ST data compared to scRNA-seq data, our pre-experiments indicate that it is difficult to converge in early training. Considering that Geneformer is one of the SOTA models, and  domain adaptation was performed using a large amount of data, we believe there will not be a significant difference if using other SOTA models such as scGPT [3].
>
> >Q4: Batch effect correction.
>
> Please refer to the response Q5 to Reviewer Rgcj.
>
> >Q5: Catastrophic forgetting.
>
> Catastrophic forgetting may cause the model's suitability for scRNA-seq data to decrease while becoming more suitable for ST data.
> - SToFM is a model specialized for ST data, and we do not recommend users to apply SToFM on scRNA-seq data scenarios.
> - We conduct an experiment on scRNA-seq data. As shown in *Rebuttal-Table 4*, there is almost no drop in performance. This may be because the distribution of gene expressions in ST data and scRNA-seq data is similar.
>
> >Q6: scGPT in Table 1.
>
> Apologies for the confusion. The table header is "ST Pretraining", and what we want to express is that scGPT did not use gene expressions from ST data during pretraining.
>
> >Q7: Leiden cannot control the number of clusters.
>
> After clustering, the number of clusters will be checked. If it does not fall within 20-100, the resolution of Leiden will be adjusted to re-cluster until the cluster number meets the requirement.
> ___
> Thank you again for your insights which are invaluable in solidifying our work. Should our responses address your queries, we would deeply appreciate your support.

---

> > ### Comment · Reviewer_tnYZ · 2025-04-02
> >
> > The response by the authors look thorough and satisfactory. Before I contemplate about changing the scores, I want authors to provide additional numbers & details if possible. I have noticed that most of the rebuttal experiments have been reported with F1.
> > - Can authors provide details behind F1 score, which I think is important, in the context of multiclass settings.
> > - Can authors provide **balanced accuracy** score (or macro-averaged AUC) as well, so we have different views of the same experiment.

---

> > > ### Author Response · Authors · 2025-04-03
> > >
> > > Thank you for your appreciation! We do our best to provide more detailed information below:
> > >
> > > >Comment Q1:  Details behind F1 score
> > >
> > > - Line 331 of the paper states that we use macro F1-score for multi-class classification problems, which means calculating the F1-score for each class and then taking the average. As follows:
> > > $F1_i=\frac{2P_iR_i}{P_i+R_i}$,
> > > macro-$F1=\frac{1}{N}\sum_iF1_i$,
> > > Where $P_i$, $R_i$, and $F1_i$ are the precision, recall, and F1 score of class $i$, respectively. N is the number of classes.
> > > - Below, as an example, we provide the intermediate results of calculating the macro F1-score of SToFM on Embryo2. As shown in *Rebuttal-Table 1*, there are 16 classes. (The results reported in the paper are average of three repeated experiments, and the results below are from one of them.)
> > > |Class|Number|Precision|Recall|F1-score|
> > > |-|-|:-:|:-:|:-:|
> > > |Brain|153|0.899|0.758|0.823|
> > > |Branchial Arch|45|0.714|0.333|0.455|
> > > |Cloaca|71|0.922|0.831|0.874|
> > > |Ganglion|131|0.782|0.519|0.624|
> > > |Heart|243|0.956|0.979|0.967|
> > > |Hepatic Diverticulum|44|0.740|0.841|0.787|
> > > |Limb Ectoderm|187|0.843|0.888|0.865|
> > > |Lung Primordium|96|0.738|0.823|0.778|
> > > |Meninges|181|0.727|0.796|0.760|
> > > |Mesonephron|51|0.864|0.745|0.800|
> > > |Pancreas Bud|54|0.920|0.852|0.885|
> > > |Pharyngeal|64|0.714|0.781|0.746|
> > > |Primitive Gut|92|0.784|0.870|0.825|
> > > |Somite|463|0.842|0.842|0.842|
> > > |Spinal Cord|422|0.877|0.943|0.909|
> > > |Surface Ectoderm|246|0.864|0.907|0.885|
> > > |**Macro F1-score**||||**0.802**|
> > >
> > > >Comment Q2: Balanced accuracy and macro-averaged AUC-ROC
> > >
> > > For experiments of Rebuttal-Table 2-4, we additionally calculate the balanced accuracy and macro-averaged AUC-ROC, which are reported below:
> > > - Rebuttal-Table 2
> > > |Data volume|Macro F1|Balanced accuracy|Macro AUC-ROC|
> > > |-|:-:|:-:|:-:|
> > > |**Embryo2**||||
> > > |12.5%|0.758|0.764|0.968|
> > > |50%|0.782|0.790|0.968|
> > > |100%|0.801|0.799|0.972|
> > > |**EmbryoCross**||||
> > > |12.5%|0.423|0.518|0.927|
> > > |50%|0.450|0.546|0.930|
> > > |100%|0.459|0.551|0.933|
> > > - Rebuttal-Table3
> > > |Model|Macro F1|Balanced accuracy|Macro AUC-ROC|
> > > |-|:-:|:-:|:-:|
> > > |**Embryo2**||||
> > > |w/o $\mathcal{L}_{PDR}$|0.749|0.756|0.965|
> > > |w/o spatial distance matrix|0.721|0.704|0.957|
> > > |SToFM|0.801|0.799|0.972|
> > > |**EmbryoCross**||||
> > > |w/o $\mathcal{L}_{PDR}$|0.437|0.520|0.929|
> > > |w/o spatial distance matrix|0.413|0.525|0.920|
> > > |SToFM|0.459|0.551|0.933|
> > > - Rebuttal-Table 4
> > > |Model|Macro F1|Balanced accuracy|Macro AUC-ROC|
> > > |-|:-:|:-:|:-:|
> > > |scBERT|0.905|0.906|0.988|
> > > |Geneformer|0.957|0.949|0.995|
> > > |SToFM-CellEncoder|0.944|0.949|0.993|
> > >
> > > We will also add more details and evaluation results in the revised edition of the paper.
> > > ___
> > > Thank you again for your help in solidifying our work!

---

### Official Review · Reviewer_LBxH · 2025-03-12

**Overall Recommendation:** 3

**Summary:**

This paper introduces a foundation model for cell spot representation of spatial transcriptomics. It fine-tunes the pretrained cell embedding (from existing scRNA foundation models) by incorporating the location information via masked feature prediction and noised distance information recovery, as well as involving the proposed virtual macro cells.   A series of down-stream tasks are performed with the pretrained spot embeddings to demonstrate the effectiveness of them.

**Claims And Evidence:**

The ablation study is not sufficient to support the effectiveness of the fine-tuning with the incorporation of distance information.

**Essential References Not Discussed:**

Most essential references are mentioned.

**Experimental Designs Or Analyses:**

The experiments are extensive and relatively comprehensive.

**Methods And Evaluation Criteria:**

For the Pairwise Distance Recovery (PDR), it is not sure that the cell embedding contains sufficient distance information to make the distance recovery mechanism feasible.

**Other Comments Or Suggestions:**

N/A

**Other Strengths And Weaknesses:**

It is an important direction to pretrain spot representation with the consideration of location information.

This paper is well written and easy to follow.

The proposed model only trained on cell resolution spatial transcriptomics data, and the feasibility of extending to low resolution ST remains unknown.

**Questions For Authors:**

Not very sure the effectiveness of the way to formulate the virtual cell, i.e., simple by averaging of the representation and location.

**Relation To Broader Scientific Literature:**

It is an important direction to develop sing-cell rna expression, and there emerge a large body of pretrained foundation model for this purpose. This paper extend to incorporate location information of ST for better spot embedding learning.

**Theoretical Claims:**

No theoretical contribution is introduced in this work.

---

> ### Author Rebuttal · Authors · 2025-03-30
>
> Dear Reviewer LBxH:
>
> Thanks for your appreciation and detailed review. We try our best to response the questions below:
>
> >Q1: The ablation study is not sufficient to support the effectiveness of the fine-tuning with the incorporation of distance information.
>
> Thank you for your suggestions! We conduct an ablation study on the spatial distance matrix. As shown in the Table below, the results demonstrate that removing the spatial distance matrix significantly decreases model performance. Essentially, the SE(2) Transformer relies on the spatial distance matrix to establish relationships between cells. If spatial distance matrix is removed, the ability of the model to perform intercellular information interactions will be compromised.
>
> |Model|Embryo2-F1| EmbryoCross-F1|
> |-|:-:|:-:|
> |w/o distance matrix|0.721|0.413|
> |SToFM|0.801|0.459|
>
> >Q2: For the Pairwise Distance Recovery (PDR), it is not sure that the cell embedding contains sufficient distance information to make the distance recovery mechanism feasible.
>
> - **Intuitively**, the spatial autocorrelation of cellular expression profiles [1], as well as the ligand-receptor intercellular signaling pathways of neighboring cells can help to recover the original distances from the noisy data.
> - **Experimentally**, the PDR loss can be converged well during our pre-training process. Moreover, following the discussion in Uni-Mol [2] about recoverability of distances, we only add limited noise to the cell coordinates, which does not obscure all spatial location information. This makes this pre-training task more feasible.
>
> >Q3: The proposed model only trained on cell resolution spatial transcriptomics data, and the feasibility of extending to low resolution ST remains unknown.
>
> - **SToFM can be applied to low-resolution ST data without domain adaptation.** In the DLPFC experiment of Section 4.2, we specifically chose the low-resolution 10x Visium data, and SToFM performed exceptionally well in this task, demonstrating that the model can be scaled to low-resolution ST data.
> - Advancements in biotechnology in recent years have continuously improved the resolution of ST data, making the analysis of high-resolution ST data a more valuable direction in the field of bioinformatics [3].
>
> >Q4: Not very sure the effectiveness of the way to formulate the virtual cell, i.e., simple by averaging of the representation and location.
>
> - **Intuitively**, the main purpose of constructing virtual cells is to provide a summary of information from global tissue sections. After clustering by combining expression embedding and location information, each cluster should contain a set of cells that are close in location and have similar expression. Thus, by averaging the embedding and position of the cells within a cluster, it is possible to say "there is a cluster of cells with similar expression at this location". We believe this is methodologically sound. The use of average or sum for simple yet effective pooling is also widely applied in fields such as graph representation learning [4].
> - **Experimentally**, our ablation experiments demonstrate that incorporating macro-scale information through virtual cells can effectively improve model performance. In addition, we calculate the similarity of expression embeddings and spatial positions of some virtual cells with each cell in the cluster, as shown in the Table below.
> |Sample ID|Pearson correlation of expression embeddings|Cosine Similarity of positions|
> |-|:-:|:-:|
> |1|0.863|0.804|
> |2|0.871|0.833|
> |3|0.893|0.853|
>
> The results indicate a high similarity between the expression embeddings and positions of the virtual cell and individual cells within the cluster.
> ___
> Thank you again for your detailed review! Your insights have been invaluable in aiding us to enhance and solidify our work. Should our responses satisfactorily address your queries, we would deeply appreciate your support for our work.
>
> Refs:
>
> [1] Mapping the transcriptome: Realizing the full potential of spatial data analysis
>
> [2] Uni-Mol: a universal 3D molecular representation learning framework
>
> [3] Methods and applications for single-cell and spatial multi-omics
>
> [4] Graph pooling in graph neural networks: methods and their applications in omics studies

---

### Official Review · Reviewer_Rgcj · 2025-03-14

**Overall Recommendation:** 3

**Summary:**

The paper proposes a multi-scale foundation model to integrate macro-scale tissue morphology, micro-scale cellular microenvironment and gene-scale gene expression profile of  spatial transcriptomics. The author constructs a large-scale spatial transcriptomics corpus containing approximately 2,000 tissue slices and 88 million cells for pretraining, which is claimed to be released. Various downstream tasks are validated to prove the performance.

**Claims And Evidence:**

The paper claims to propose a multi-scale foundation model that captures and integrates information from macro, micro and gene scale of spatial transcriptomics. The ablation on the three scale is conducted in Table 5. Large performance improvement is shown at the second and fourth row in Tabel 5. However, performance improvement involving micro scale by adding the SE(2) Transformer is relatively limited. The author should include an ablation experiment evaluating the model’s performance when using only macro- and gene-scale features without micro-scale components. Besides, the author requires isolating the effect of the spatial distance matrix through experiments comparing performance with and without this component. The sensitivity of sample rate on the second-time cell encoding mentioned in Sec. 3.2 should be explored. And the effect of incorporating spatial information should also be evaluated in ablation study.

**Essential References Not Discussed:**

No

**Experimental Designs Or Analyses:**

The experimental designs evaluates the model’s performance in various downstream tasks. The experiments on tissue region semantic segmentation evaluate the model’s ability to understand the functional specialization of cells. The author claims the experiments on cell type annotation in spatial transcriptomics confirm that incorporating spatial information can help improve cell type annotation. But the effect of incorporating spatial information should be evaluated in ablation study. Experiments on zero-shot clustering and visualization illustrate the ability of the model to produce high-quality cell embeddings.The model is also is evaluated to be a effective tool for spatial deconvolution. Experiments on spatial transcriptomics imputation shows the model’s ability to infer the uncaptured gene expression levels. The paper claims to propose a multi-scale foundation model that captures and integrates information from macro, micro and gene scale of spatial transcriptomics. The ablation on the three scale is conducted in Table 5. Large performance improvement is shown at the sencond and fourth row in Tabel 5. However, performance improvement involving micro scale by adding the SE(2) Transformer is relatively limited. It’s needed to evaluate what’s the performance when the model involve both Gene and Macro, except for Micro. Besides, the ablation on the effect of the spatial distance matrix is also needed to be evaluated. Also, ablation on the second-time cell encoding mentioned in Sec. 3.2 is needed to be explored. How does the sample rate influence the performance? The effect of incorporating spatial information should also be evaluated in ablation study.

**Methods And Evaluation Criteria:**

The evaluation criteria makes sense for the analysis of spatial transcriptomics representation. The downstream tasks about human embryonic structure segmentation, DLPFC layer segmentation, cell type annotation in spatial transcriptomics are validated in the experiments. Accuracy and F1 are reported to evaluate the segmentation and cell type annotation performance. Besides, the performance in Zero-shot Clustering and Visualization and Spatial Deconvolution illustrates the model’s ability  to produce high-quality cell embeddings.

**Other Comments Or Suggestions:**

No

**Other Strengths And Weaknesses:**

This paper is clearly written and easy to read. In terms of spatial transcriptome characterization methods, this thesis incorporates macro-, micro-, and gene-scale information and adds positional information, for which the largest spatial transcriptome dataset to date has been created. And the downstream task experiments in this paper are very rich. But the ablation experiment needs to be improved.

**Questions For Authors:**

1. The author should include an ablation experiment evaluating the model’s performance when using only macro- and gene-scale features without micro-scale components.
2. The author requires isolating the effect of the spatial distance matrix through experiments comparing performance with and without this component.
3. The details on the second-time cell encoding mentioned in Sec. 3.2 should be written. If a sample rate is involved in the second-time cell encoding, the sensitivity of sample rate should be explored.
4. The effect of incorporating spatial information should also be evaluated in ablation study.
5. This paper migrates the single-cell model through a domain adaptation strategy. However, there may be relatively large differences between different ST datasets, such as different technical platforms. How should we further improve adaptability?

**Relation To Broader Scientific Literature:**

First, the author we construct SToCorpus-88M, the largest high-resolution ST pretraining corpus to date, which contains approximately 2,000 tissue slices and 88 million cells and is claimed to be released. This contribution will provide a large dataset for further ST analysis.  Second, the author proposes a multi-scale foundation model to integrate macro-scale tissue morphology, micro-scale cellular microenvironment and gene-scale gene expression profile of spatial transcriptomics, which provides a idea for integrate multi-scale information from ST.

**Theoretical Claims:**

No

---

> ### Author Rebuttal · Authors · 2025-03-30
>
> Dear Reviewer Rgcj:
>
> Thanks for your appreciation and detailed review. We try our best to response the questions below:
> >Q1: Ablation experiment on micro-scale components. **&** Q2: The effect of the spatial distance matrix. **&** Q4: The effect of incorporating spatial information.
>
> Thank you for your suggestions for refining the ablation experiments!
> - For Q1, We ablate the micro-scale by limiting the scale of sub-slices to 1, to ensure that the cells could only interact informationally with the virtual cells that represent macro-scale information, but not with other cells in the micro-environment. Experimental results are shown in the Table below. A significant decrease in performance is observed, which demonstrates the effectiveness of incorporating the micro-scale information.
> - For Q2, we conducted an ablation study on the spatial distance matrix. The spatial information is very underutilized in this case, only used to construct virtual cells and divide sub-slices.  As shown in the Table below, the results demonstrate that removing the spatial distance matrix significantly decreases model performance.
> - For Q4, we had presented the results of completely ablating spatial information in the second row "Cell Encoder w/ DA" of Table 5 of our paper. We repeat the results in the Table below. Essentially, we rely on spatial information to establish relationships between multiple cells, with both micro- and macro-scale information being part of spatial information. If spatial information is removed, the model will not be able to interact between cells at all, essentially degrading to independently encoding each single cell.
>
> |Model|Embryo2-F1| EmbryoCross-F1|
> |-|:-:|:-:|
> |Q1: w/o micro|0.721|0.425|
> |Q2: w/o spatial matrix|0.721|0.413|
> |Q4: w/o spatial information (i.e. Cell Encoder w/ DA)|0.718|0.415|
> |SToFM|0.801|0.459|
>
> >Q3: The details on the second-time cell encoding mentioned in Sec. 3.2 should be written. If a sample rate is involved in the second-time cell encoding, the sensitivity of sample rate should be explored.
> - We will present more details in the revised paper, and will release detailed code to help readers better understand and reproduce our approach.
> - We will add experiments and discussion on the sample rate. The purpose of the second time cell encoding is to enable L_MCM and L_PDR to optimize the cell encoder through backpropagation. To balance the training cost and model performance, we use only a small number of cells for this computation, which is similar to selecting a smaller batch size to update the cell encoder. For the selection of the sampling number, we determined the number of samples to be 12, given that the original Geneformer paper gave a training batch size of 12.
> - Considering the computational cost, we test the impact of the sample number on a small amount of data (1/8 of the SToCorpus-88M), as shown in the Table below. The results show that the model has some robustness to this hyperparameter, just as the batch size often only affects the convergence speed rather than the model performance. However, setting the sample size to 0, i.e. freezing the cell encoder, will lead to a decrease in model performance.
> |Sample number|Embryo2-F1| EmbryoCross-F1|
> |-|:-:|:-:|
> |0|0.722|0.417|
> |4|0.754|0.424|
> |12|0.758|0.423|
>
> >Q5: This paper migrates the single-cell model through a domain adaptation strategy. However, there may be relatively large differences between different ST datasets, such as different technical platforms. How should we further improve adaptability?
>
> - Research such as scGPT [1] and LangCell [2] has already proven that large-scale pretraining is one of the best ways to remove batch effects in scRNA-seq data. Nicheformer [3] has also demonstrated that by pretraining, the model can gain modeling capabilities across different ST technology platforms. Indeed, despite the gap between different datasets, the underlying gene co-expression, intercellular signaling pathways, and other information should be largely uniform, which makes them have similar distributions that can be captured by the pre-trained model.
> - In the DLPFC experiments in Section 4.2, we purposely chose 10x Visium data that was not used in the pretraining. The excellent performance of SToFM on this task also proves that the model has gained the ability to transfer across technology platforms from the pretraining. We believe that our model and dataset can be of great help for future research in the application scenario of batch integration.
> ___
> Thank you again for your detailed review! Your insights have been invaluable in aiding us to solidify our work. Should our responses address your queries, we would deeply appreciate your support for our work.
>
> Refs:
>
> [1]  scGPT: toward building a foundation model for single-cell multi-omics using generative AI
>
> [2] LangCell: language-cell pre-training for cell identity understanding
>
> [3] Nicheformer: a foundation model for single-cell and spatial omics

---

> > ### Comment · Reviewer_Rgcj · 2025-04-03
> >
> > In the author's "Ablation Study", it states, "we ablate the model’s ability to jointly model multiple cells at the micro-scale by removing the SE(2) Transformer." This implies that the SE(2) Transformer is responsible for capturing micro-scale features. Given this, for Q1, wouldn't it be more appropriate to remove the SE(2) Transformer while retaining the virtual cell to assess the model’s performance using only macro- and gene-scale features, without micro-scale components? However, in the current ablation, the author limits the sub-slice scale to 1 to remove micro-scale modeling. In this case, wouldn’t the SE(2) Transformer still capture interactions among micro-scale cells within the sub-slice?

---

> > > ### Author Response · Authors · 2025-04-03
> > >
> > > Apologies for the confusion. We try our best to clarify as follows:
> > >
> > > 1. First of all, we would like to clarify that "*the SE(2) Transformer is only responsible for capturing micro-scale features*" is a misunderstanding. The SE(2) Transformer is used for encoding **both macro-scale and micro-scale** information. It captures micro-scale information by modeling intercellular relationships in the microenvironment, and captures macro-scale information by modeling relationships between cells and virtual cells. Removing SE(2) Transformer will remove both macro-scale and micro-scale information simultaneously.
> > >
> > > 2. The context of this sentence in the "Ablation Study" is: "*We **first ablate the macro-scale** information by removing the virtual cells. **Then, we ablate ... micro scale** by removing the SE(2) Transformer.*"  What we want to express is that in the case of **already ablating the macro scale**, removing the SE(2) Transformer  can **further ablate the micro scale**, i.e., ablate both macro- and micro-scale, as shown in Table 5. We will use a clearer statement in the revised paper.
> > >
> > > 3. **Q:** Wouldn't it be more appropriate to remove the SE(2) Transformer while retaining the virtual cell?
> > > **A:** We use SE(2) Transformer to integrate information from cells and virtual cells. Therefore, it is unreasonable to conduct an ablation study that remove SE(2) Transformer but retain virtual cells.
> > >
> > > 4. **Q:** The author limits the sub-slice scale to 1 to remove micro-scale modeling. In this case, wouldn’t the SE(2) Transformer still capture interactions among micro-scale cells within the sub-slice?
> > > **A:** SE(2) Transformer captures micro-scale information by modeling cells in the microenvironment, and captures macro-scale information by modeling virtual cells. Therefore, we remove the cellular microenvironment by setting the sub-slice size to 1 in rebuttal Q1. In this case, only one cell and some virtual cells are input to the SE(2) Transformer. **Since there is only one cell in the sub-slice, the microscale information is de facto absent, and the SE(2) Transformer is naturally unable to capture the micro-scale cell-cell interactions.**
> > >
> > > 5. More intuitively, We summarize the relationship between modules and scales as follows:
> > > **Virtual Cells + SE(2) Transformer -> macro-scale**
> > > **Microenvironment + SE(2) Transformer -> micro-scale**
> > > |Ablation|gene scale|micro scale|macro scale|
> > > |-|:-:|:-:|:-:|
> > > |w/o SE(2) Transformer (*CellEncoder w/ DA* in **Table 5**)|$\checkmark$|$\times$|$\times$|
> > > |w/o virtual cells (*SToFM w/o VCs* in **Table 5**)|$\checkmark$|$\checkmark$|$\times$|
> > > |w/o microenvironment (*w/o micro* in **Rebuttal Q1**)|$\checkmark$|$\times$|$\checkmark$|
> > >
> > > ___
> > > Thank you again for your help in solidifying our work!

---

### Official Review · Reviewer_TZDp · 2025-03-14

**Overall Recommendation:** 2

**Summary:**

The paper proposes SToFM, a multi-scale Spatial Transcriptomics foundation model, to effectively integrate macro-, micro-, and gene-scale information from Spatial Transcriptomics (ST) data. SToFM uses a combination of gene expression profiles, cell coordinates, and spatial relationships to learn representations of cells in their tissue context. It employs domain adaptation for gene expression embeddings, integrates spatial information using an SE(2) Transformer, and introduces novel pretraining tasks like masked cell modeling and pairwise distance recovery. The model outperforms existing methods on several biological tasks, demonstrating its ability to capture complex multi-scale biological information.

**Claims And Evidence:**

There may be a domain gap between ST data and scRNA-seq data. While ST data contains spatial information, scRNA-seq data only includes gene expression values. Therefore, the justification for bridging the significant gap between these two data types through transfer learning is insufficient.

**Essential References Not Discussed:**

No

**Experimental Designs Or Analyses:**

The alpha value in Algorithm 1 represents the combination ratio of cell embeddings and cell positions. However, the paper lacks an analysis of the model's sensitivity to different alpha values. It is necessary to examine how varying alpha values impact the integration of multi-scale information.

The impact of the combination ratio of the two loss functions, L_MCM and L_PDR, on the model's performance should be analyzed.

**Methods And Evaluation Criteria:**

Since this model requires pretraining of the cell encoder, its scalability is limited. Additionally, training both the cell encoder and the SE(2) Transformer can result in significant computational costs. However, the paper lacks running time experiments and time complexity analysis.

**Other Comments Or Suggestions:**

No other Comments

**Other Strengths And Weaknesses:**

No other Strengths And Weaknesses

**Questions For Authors:**

Please refer to the above contents.

**Relation To Broader Scientific Literature:**

The main contribution of this paper is the integration of multi-scale information from Spatial Transcriptomics (ST) data, including gene expression, cellular interactions, and tissue morphology. Unlike prior models, SToFM effectively combines micro-scale (cellular) and macro-scale (tissue) information using a multi-scale approach and SE(2) Transformer. This approach captures richer, more comprehensive biological insights from ST data.

**Theoretical Claims:**

This paper does not present theoretical analysis.

---

> ### Author Rebuttal · Authors · 2025-03-29
>
> Dear Reviewer TZDp:
>
> Thanks for your appreciation and detailed review. We try our best to response the questions below:
> >Q1: Bridging the gap between ST and scRNA-seq data through transfer learning.
>
> ST data consists of two parts: **spatial location** and **gene expression values**. As SToFM is a model for ST data, the purpose of domain adaptation is to utilize well-trained scRNA-seq model to better encode the **gene expression values** of ST data. The gene expression values of ST data have a similar distribution to scRNA-seq data and follow the same underlying gene co-expression patterns. Many well-established bioinformatics methods have also proven the effectiveness of transferring knowledge between scRNA-seq and ST data, such as Tangram[1]. Therefore, we believe it is reasonable to apply transfer learning between  scRNA-seq data and the **gene expression values** of ST data.
>
> >Q2: Since this model requires pretraining of the cell encoder, its scalability is limited.
>
> Research like scGPT[2] has already shown that Transformer-based cell encoder models have good scalability. We believe that a well-pretrained cell encoder can help improve the model's scalability.
> >Q3: Computational costs. Lack of running time experiments and time complexity analysis.
>
> - The cell encoder is based on the Transformer architecture. And the time complexity of the SE(2) Transformer is similar to that of a standard Transformer [3]. Therefore, the time complexity of both parts of SToFM is **the same as the standard Transformer**. Specifically, the time complexity is $O(N *n^2+M *m^2)$, where $N$ and $n$ are the number of cells and genes, and $M$ and $m$ are the number and the scale of the sub-slices, respectively.
> - Section 4.2 of our paper provides details on the pretraining time cost. We have also added runtime experiments for inference, as shown in the Table. Specifically, running inference on an ST slice containing tens of thousands of cells typically takes 1–5 minutes, which we consider acceptable for practical applications.
>
> |Slice|Gene number|Cell number|Sub-slice number|Cell encoder time (s)|SE(2) Transformer time (s) | Total time (s)
> |-|-|-|-|:-:|:-:|:-:|
> |Allen1(Sec. 4.4)|355|21002|23|28.6|21.1|49.7|
> |Embryo2(Sec.4.2)|17552|12865|14|211.5|15.9|217.4|
>
> >Q4: Model's sensitivity to different alpha values.
>
> Thank you for your suggestion! We have added experiments to show how different alpha values affect the model’s performance, as shown in the Table below:
> |alpha|Embryo2-F1| EmbryoCross-F1|
> |-|:-:|:-:|
> |0|0.751|0.435|
> |0.2|0.767|0.458|
> |0.4|0.778|0.440|
> |0.6|0.796|0.436|
> |0.8|**0.801**|**0.459**|
> |1.0|0.769|0.448|
>
> The model has a certain robustness in alpha, and we believe this may be because cells that are closer in location are more likely to have similar gene expressions [4]. The alpha=0.8 that we chose is essentially the optimal setting.
>
> >Q5: The impact of the combination ratio of the two loss functions.
>
> We will introduce further analysis on the loss ratio in the revised version of the paper. These two loss are relatively close in scale, and in our experiments, combining them in a 1:1 ratio allows both to converge normally. Considering the computational cost, we test the impact of the ratio of the two losses on the speed of convergence and the performance of the model on a small amount of data (1/8 of the SToCorpus-88M), as shown in Rebuttal-Fig.1 (https://anonymous.4open.science/api/repo/stofm-rebuttal/file/rebuttal-TZDp.pdf?v=5b2ab9f4) and the Table below. ($\gamma$ is the loss ratio in $L=\gamma*L_{MCM} + (1- \gamma) * L_{PDR}$)
>
> |$\gamma$|Embryo2-F1| EmbryoCross-F1|
> |-|:-:|:-:|
> |0|0.683|0.409|
> |0.2|0.729|0.423|
> |0.5|**0.758**|0.423|
> |0.8|0.753|**0.429**|
> |1.0|0.701|0.415|
>
> The results show that the model's performance is robust to the ratio of loss functions. However, removing one of the loss functions will lead to a decrease in model performance. The $\gamma$=0.5 that we chose is essentially the optimal setting.
>
> >Q6: The provided code does not run.
>
> We had uploaded the code of the model and core algorithms along with the paper to help better understand the method. The executable code has been organized now, but anonymous links containing code are not allowed during rebuttal. We promise to release the executable code on github after the paper is published.
> ___
> Thank you again for your detailed review! Your insights are invaluable in aiding us to solidify our work. Should our responses satisfactorily address your queries, we would deeply appreciate your support for our work.
>
> Refs:
>
> [1] Deep learning and alignment of spatially resolved single-cell transcriptomes with Tangram
>
> [2] scGPT: toward building a foundation model for single-cell multi-omics using generative AI
>
> [3] Uni-Mol: a universal 3D molecular representation learning framework
>
> [4] Mapping the transcriptome: Realizing the full potential of spatial data analysis

---

> > ### Comment · Reviewer_TZDp · 2025-04-08
> >
> > 1. It is necessary to specifically justify how the gap between spatial information and gene expression can be bridged. Additional experiments, such as visualization, are required to demonstrate the validity of the transfer learning approach.
> >
> > 2. To prove the efficiency regarding the pre-training time cost, it is essential to compare the running time with various other baselines. However, the authors have only presented the running time of the proposed method.
> >
> > 3. As the complete code is not available, I was unable to attempt reproduction of the provided experimental results. This poses a problem in validating the reliability of the model.
> >
> > I will maintain my current score.

---

> > > ### Author Response · Authors · 2025-04-08
> > >
> > > Dear Reviewer TZDp,
> > >
> > > Thanks for your detailed comments. We try our best to address any further concerns:
> > >
> > > > 1. It is necessary to specifically justify how the gap between spatial information and gene expression can be bridged. Additional experiments, such as visualization, are required to demonstrate the validity of the transfer learning approach.
> > >
> > > - We would like to clarify that, as mentioned in line 182-205 of the paper and Rebuttal Q1, our method does not perform transfer learning between spatial information and gene expression. Instead, we focus on transfer learning between gene expression data from ST and scRNA-seq.
> > > - It is widely accepted in the bioinformatics community that gene expression profiles in ST and scRNA-seq data tend to have similar distributions. For example, when deconvolution is performed on ST data, the scRNA-seq dataset is often used as the reference dataset, as detailed in the benchmark paper [1].
> > > - We have added a visualization experiment, as shown in the anonymous link https://anonymous.4open.science/api/repo/icml-reply-2256/file/reply-TZDp.pdf?v=9b10dd35. In this experiment, we use three types of cells from the mouse brain, which were obtained from an ST slice (Allen1 in Sec. 4.4) and a scRNA-seq dataset [2]. We demonstrate the UMAP visualization of both the original expression levels and the cell embeddings obtained using the cell encoder of SToFM. The results shows that the gene expression of different cell types followed a similar relative distribution between the ST data and the scRNA-seq data (*Reply-Figure 1*). Furthermore, through domain adaptation, the cell encoder of SToFM is able to further bridge the data gap (*Reply-Figure 2*).
> > >
> > > > 2. To prove the efficiency regarding the pre-training time cost, it is essential to compare the running time with various other baselines. However, the authors have only presented the running time of the proposed method.
> > >
> > > Thank you for your suggestion. We have provided the running times of different models using two slices in Rebuttal Q3 as examples, in the table below. All experiments, except PCA, were conducted on a single NVIDIA Tesla A100 GPU. We found that most models are able to complete the calculation for an ST slice in a few minutes, which we believe is a reasonable processing time for practical applications in single-cell analysis. The computational efficiency of SToFM is similar to that of other Transformer-based models for processing gene sequences (Geneformer, Nicheformer) when dealing with large slices like Embryo2.
> > >
> > > |Slice|Allen1(Sec. 4.4)|Embryo2(Sec. 4.2)|
> > > |-|:-:|:-:|
> > > |Gene number|355|17552|
> > > |Cell number|21002|12865|
> > > |PCA (s)|16.3|229.6|
> > > |Geneformer (s)|29.3|208.0|
> > > |Nicheformer (s)|326.9|344.45|
> > > |CellPLM (s)|41.9|76.0|
> > > |SToFM (s)|49.7|217.4|
> > >
> > >
> > > > 3. As the complete code is not available, I was unable to attempt reproduction of the provided experimental results. This poses a problem in validating the reliability of the model.
> > >
> > > We understand the importance of code availability for reproducibility and are committed to ensuring that all necessary resources are accessible after the anonymous review. Additionally, our supplementary materials contain detailed code implementations of all the core methods in the paper. The addition of checkpoint, data and simple script files would be sufficient for execution. We kindly request that this be taken into account.
> > >
> > > ___
> > > Thank you again for your help in solidifying our work! If you have further concerns, please let us know by editing the Rebuttal Comment.
> > >
> > > Refs:
> > >
> > > [1] Benchmarking spatial and single-cell transcriptomics integration methods for transcript distribution prediction and cell type deconvolution
> > >
> > > [2] Thyroid hormone remodels cortex to coordinate body-wide metabolism and exploration

---

### Decision · Program_Chairs · 2025-05-01

**Decision:**

Accept (poster)

**Comment:**

Following the recent trend of foundation models, the paper proposes one for spatial transcriptomics. While many of the components are put together from existing work, the reviewers identify two main contributions of the paper: (1) from a technical point-of-view, incorporating distance information and (2) a data contribution, the new SToCorpus-88M dataset. The reviewers initially raised several concerns about missing empirical experiments and ablation studies but these were addressed by the authors during the course of the rebuttal period. As a result, I believe the paper makes a solid contribution to the conference and should be accepted.